# Sugarcane Bagasse and Orange Peels as Low-Cost Biosorbents for the Removal of Lead Ions from Contaminated Water Samples

**Ntsieni Romani Molaudzi and Abayneh Ataro Ambushe ***

Department of Chemical Sciences, University of Johannesburg, P.O. Box 524, Johannesburg 2006, South Africa
* Correspondence: aambushe@uj.ac.za

**Abstract:** The pollution of water by lead (Pb(II)) constitutes a substantial threat to the environment and subsequently to animals and humans. In this study, the efficacy of sugarcane bagasse (SCB) and orange peels (OPS) have been investigated as potential low-cost biosorbents, individually and in a homogeneous combination for the removal of Pb(II) from simulated and real water samples. Biosorbents were characterised using Fourier transform infrared (FTIR) spectroscopy, thermogravimetric analysis (TGA), scanning electron microscopy (SEM) coupled to energy-dispersive X-ray spectroscopy (EDS), transmission electron microscopy (TEM), powder X-ray diffraction (pXRD), a zeta potential analyser, and the Brunauer–Emmett–Teller (BET) method. Batch adsorption studies were explored under several experimental conditions to optimise the removal efficiency of Pb(II) ions from artificially contaminated aqueous solutions. The pH study revealed optimum removal efficiencies of Pb(II) at pH 7, for SCB and OPS. The optimum contact time for SCB and OPS individually and a homogenous mixture was 60, 120, and 120 min, respectively. The study also revealed that the optimum biosorbent dosage was 0.2, 0.17, and 0.2 g for SCB, OPS, and the homogenous combination of SCB and OPS (1:1). Optimum experimental conditions could achieve up to 100% removal efficiencies for 10 and 20 mg/L of Pb(II) using SCB and OPS, respectively. The potential of the homogenised combination of biosorbents demonstrated 100% removal efficiencies for 10 mg/L of Pb(II). The removal of 10 mg/L of Pb(II) in real water samples remained at 100% for biosorbents individually and the homogenised combination. The reusability performance of SCB, OPS, and the homogenised combination of SCB and OPS presented Pb(II) removal efficiencies above 70% for three adsorption–desorption cycles.

**Keywords:** sugarcane bagasse; orange peels; biosorbent; lead; removal efficiency; adsorption–desorption

## 1. Introduction

The deterioration of water quality and quantity has been identified as a potential threat to the socio-economic development of South Africa. Although South Africa is a semi-arid country, the depreciation in the quality and quantity of water has been highly attributed to salinisation, eutrophication, sedimentation, commercial agriculture, water contamination by pathogens, pesticides, potentially toxic elements (PTEs), and other pollutants from built-up areas [1]. Although the natural dissemination of water pollutants into the environment occurs by geological and biological activities, the contribution of anthropogenic activities to water pollution is substantial. The civilisation of humans has always been dependent on the ability to find and use safe and clean water. This has been the case since humans began to find settlement in areas invariably close to water sources such as groundwater reservoirs, rivers, and lakes [2,3].

Lead (Pb) is the 82nd most prevalent element in the crust of the earth. It typically exists as a sulphur-bearing compound (PbS) in the earth's crust. As compared to many metallic species, the natural occurrence of Pb in the earth's crust and surface is infrequent [4].

However, Pb is one of the most ubiquitous water contaminants. The prevalence of Pb in the ecosystem is an environmental concern. Lead manifests in several foods, the atmosphere, water, and soil [5–7].

Lead is categorised amongst the most toxic PTEs. It is a non-biodegradable metal that has no health benefits. The consumption of Pb by plants, humans, and the aquatic ecosystem can be highly toxic [8,9]. In humans, PTEs such as Pb(II) are accumulated in the body by the ingestion of contaminated food and drinking water or inhaling polluted air. The persistent, toxic, and non-biodegradable nature and exposure to Pb(II) at high concentrations may result in cancer, an increase in blood pressure, congenital disabilities, anaemia, mental retardation, and severe kidney and brain damage [8,10]. It is to this effect that regulatory organisations such as World Health Organisation (WHO) and South African National Standards (SANS) set a maximum permissible level (MPL) of 0.01 mg/L in drinking water [11,12]. The pre-eminent route of entry for Pb(II) in water systems is through anthropogenic activities such as pipelines' corrosion and industrial activities such as metal processing, printing, mining, brewing using Pb-coated tanks, glass production, petroleum refinery, metal mining, and the manufacturing of batteries and explosives [5,8,13–16]. In South Africa, concentrations of Pb that exceed the MPL set by SANS and the WHO were reported in study areas such as the Mokolo River, Groot and Klein Nyl River, Dzindi River, and Umgeni River. The study areas were found invariably in proximity to anthropogenic activities such as farming, manufacturing of multiple commodities, and mining [17–20].

Over the years, to combat the effect of humans' and animals' exposure to water contaminated by PTEs such as Pb(II), water treatment techniques such as filtration, ion exchange, precipitation, reverse osmosis, flocculation, coagulation, oxidation, and adsorption have been employed [21]. Amongst the mentioned water treatment technologies, adsorption is considered significantly superior. The application of adsorption has fewer limitations as it is much easier to operate, has an outstandingly simple design, and has reduced sludge production and disposal costs compared to conventional water treatment techniques [22]. Adsorption using activated carbon (AC) is the most studied technique. Adsorption using AC is presumably an ideal technique considering that it complies with most preconditions of a suitable water treatment technique. However, the cost of production of AC is economically strenuous for the treatment of large amounts of water [23].

In the past decade, it has come to the attention of water treatment scientists and engineers that there is a need to design and adopt effective, sustainable, safe, cheap, time-efficient, and environmentally friendly water treatment technologies [24,25]. A combination of these properties suits the criteria of an ideal water treatment technique. Although the extraction of selected PTEs by conventional water treatment technologies remains highly efficient, these technologies often do not meet the criteria. Conventional water treatment techniques often require large amounts of energy and costly materials and methods [23,24,26]. The search for an adsorption-based water remediation technique that satisfies the criteria of a suitable treatment technology for the removal of Pb(II) in contaminated water has been investigated in this study.

The application of agricultural by-products as potential adsorbents in wastewater treatment is not only economically feasible but is an excellent representation of applicable green chemistry due to the ease of operation, handling, and storing [23,24]. Multiple agricultural industries produce large amounts of agricultural waste without significant secondary application. Employing these agricultural by-products in wastewater treatment will create a noble destination for the organic load that has been deemed agriculturally futile [27–30]. Potential agricultural by-product adsorbents encompass waste materials such as sugarcane bagasse (SCB), coconut shells, rice husks, sawdust, maize cob, and orange peels (OPS) [23,24].

In this study, SCB and OPS are explored as low-cost green biosorbents for their potential in the remediation of water contaminated by toxic Pb(II). Therefore, it is imperative to search for a low-cost and sustainable biosorbent for decontamination of this toxic metal ion from water. To the best of our knowledge, the use of a homogenised combination of

SCB and OPS for the removal of Pb(II) from contaminated water has been investigated for the first time, in this study. The use of a homogenised combination of SCB and OPS is important to avoid seasonal-scarcity of biosorbents that are fit for the removal of Pb(II) from wastewater. This study used a wide range of analytical techniques for characterisation of SCB and OPS. The experimental parameters that influence adsorption of Pb(II) such as pH of solution, contact time, adsorbent dosage, and initial metal ion concentration in artificially contaminated aqueous solutions were investigated. Finally, decontamination of Pb(II) from spiked borehole water containing high concentrations of cations and anions such as calcium, magnesium, chloride, and sulphate ions was also investigated at optimised experimental conditions.

## 2. Materials and Methods

### 2.1. Reagents and Standards

Analytical grade, commercially produced $Pb(NO_3)_2$ 99.0% (Sigma-Aldrich, St. Louis, MO, USA) was utilised for the preparation of simulated aqueous solutions and spiking of real water samples. A 65% Suprapur $HNO_3$ (Merck, Darmstadt, Germany) and 98% NaOH pellets (Merck, Germiston, South Africa) were used to adjust the pH level. A 96% ethanol (Thembane Labs, Johannesburg, South Africa) was used to prepare samples for characterisation by transmission electron microscopy (TEM). Ultrapure deionised water from a MilliQ Direct 8 Water Purifier System (Millipore S.A.S., Molsheim, France) with a resistivity of 18.2 MΩ·cm at 25 °C was used for the preparation of all solutions. The calibration standards were prepared using 1000 mg/L of Pb (Sigma-Aldrich, St. Louis, MO, USA). Acetylene ($C_2H_2$) 98% (AFROX, Johannesburg, South Africa) and air (AFROX, Johannesburg, South Africa) were utilised to produce an Air-$C_2H_2$ flame type in flame atomic adsorption spectrometry (FAAS).

### 2.2. Instrumentation and Equipment

An XS pH meter (XS instruments, Carpi, MO, Italy) was used to measure the pH of all solutions, an FMH magnetic stirrer (FMH laboratory products, Alberton, South Africa) was used to agitate biosorbents in solution, and a 3 L laboratory ultrasonic cleaning bath (MRC Laboratory-Instruments, Holon, Israel) was used to sonicate solutions for zeta potential and TEM. Biosorbents were ground using a Nima electric grinder (Nima YK, Osaka, Japan), and all glassware was dried using a 310 R01 Glassware Drier (LABOTECH, Midrand, South Africa). Characterisation of biosorbents was conducted using a Malvern Zetasizer (Malvern Panalytical, Randburg, South Africa) to determine the influence of pH on surface charge. Biosorbent surface functional groups were determined using an IR Affinity-1S FTIR Spectrophotometer (Shimadzu, Kyoto, Japan), and a PerkinElmer STA 6000 thermogravimetric analyser (TGA) (TA Instruments, New Castle, DE, USA) was used to investigate the thermal stability of biosorbents. A TESCAN VEGA 3 LMH scanning electron microscope (SEM) (TESCAN, Brno-Kohoutovice, Czech Republic) coupled to an energy-dispersive spectrometer (Oxford Instruments, Buckinghamshire, UK) and a JOEL JEM 2100F transmission electron microscope (Joel, Tokyo, Japan) were used to determine the surface morphology of SCB and OPS. The crystallographic arrangement of biosorbents was examined using a PANanalytical Philips X'Pert Pro Powdered X-ray diffractometer (PANanalytical, Almelo, The Netherlands). A Mircromeritics ASAP 2020 gas absorption Brunauer–Emmett–Teller (BET) (Micrometrics, Brussels, Belgium) was used to determine the adsorbents' surface area, pore volume, and average pore size diameter. Concentrations of Pb(II) in all samples were quantified using a Shimadzu AA 7000 FAAS (Shimadzu, Kyoto, Japan).

### 2.3. Biosorbents' Collection and Preparation

Sugarcane was purchased from a street market in Thoyondou, Limpopo province. Oranges were purchased at a Pick n Pay in Johannesburg, Gauteng province. The hard exterior of sugarcane was peeled off, and the juice was extracted from the fibular part of

sugarcane. The dry pulpy fibrous remnants were then kept for later usage. The OPS were obtained by peeling oranges. The SCB and OPS were washed with tap water, then with deionised water to remove dust particles. The biosorbents were air-dried in a clean room for 7–10 days. The dried adsorbents were ground using an electric grinder and sieved using a sieve with a pore size of 5 μm to obtain particles of uniform size. The resulting powdered adsorbents were stored in an airtight container for later use in characterisation and batch adsorption studies.

*2.4. Characterisation Studies*

2.4.1. Characterisation by Zeta Potential Analyser

The charge on the surface of the adsorbents was determined at pH 1, 2, 4, 6, 8, 10, and 12. The surface charge of each biosorbent was determined by dissolving 0.015 g of each sorbent material in 50 mL of deionised water at their respective pH levels. Once the samples had been sonicated for an hour, appropriate amounts of the samples were transferred into a zetasizer dip cell, and the zeta potential of the adsorbents was determined at pH 1, 2, 4, 6, 8, 10, and 12.

2.4.2. Characterisation by Fourier Transform Infrared Spectroscopy

A small amount of the adsorbent was placed into a sample holder, compressed, and analysed in the wavenumber range of 400 to 4000 cm$^{-1}$. Each adsorbent was scanned 32 times at a resolution of 16 cm$^{-1}$. Samples were analysed without further preparation since an ATR probe was used. The spectra obtained for SCB and OPS were compared to other spectra on the LabSolutions IR library database.

2.4.3. Characterisation by Thermogravimetric Analysis

An aliquot of the adsorbent material was introduced into the thermogravimetric analysis (TGA) sample holder. The thermal stability of the sorbent material was assessed in the range of 25 to 1000 °C. The amount of weight lost was recorded as a function of temperature increments.

2.4.4. Characterisation by Scanning Electron Microscopy-Energy Dispersive Spectroscopy

Small amounts of adsorbents were mounted onto carbon tape and coated using an Agar Turbo Carbon Coater. To determine the morphological structures of the coated samples, a scanning electron microscope (SEM) and Vega TC software were employed at a working distance of 15 mm. The desired image of the adsorbents was generated by the magnification and focusing of the microscope. A SEM coupled to energy dispersive spectroscopy (EDS) was used to determine the elemental composition of SCB and OPS.

2.4.5. Characterisation by Transmission Electron Microscopy

Transmission electron microscopy (TEM) was used to produce more detailed information of the adsorbents' internal structure and the morphology of the surface of SCB and OPS. Before analysing the adsorbents using a TEM, a small amount of the biosorbents was sonicated in ethanol for an hour. The samples were then dispensed onto carbon-coated copper grids using a Pasteur pipette and analysed by TEM at 200 kV.

2.4.6. Characterisation by Powder X-ray Diffraction

To determine whether the adsorbents were amorphous or crystalline, aliquots of pulverised samples of the biosorbents were analysed by powder X-ray diffraction (pXRD). The biosorbents were analysed at diffraction angles (2θ) between 0 and 90° using Cu-Kα as a radiation source at 8.04 keV and at a wavelength of 1.5406 nm. The pXRD was operated at 40 mA and 40 kV to generate diffraction patterns. The pXRD diffraction patterns were then used to determine components of the adsorbent.

2.4.7. Characterisation by Brunauer–Emmett–Teller

The Brunauer–Emmett–Teller (BET) technique was used to determine the surface area, pore diameter, and pore volume. Before analysis with BET, approximately 0.3 g of the biosorbents were added into a BET tube and degassed using the $N_2$ micrometrics degassing system at 90 °C for 12 h. The surface area and pore size were determined at −195.8 °C.

*2.5. Preparation of Samples*
2.5.1. Preparation of Pb(II) Adsorbate Solutions

A 1000 mg/L stock solution of Pb(II) was prepared by dissolving 0.1599 g of $Pb(NO_3)_2$ in a 100 mL volumetric flask with deionised water. A working solution of 50 mg/L was prepared by diluting the stock solution of Pb(II) 20×. All simulated and real water samples used for batch adsorption studies were prepared in 100 mL volumetric flasks.

2.5.2. Preparation of Calibration Standards for Analysis by FAAS

Using a stock 1000 mg/L Pb standard, a 100 mg/L intermediate standard was prepared. Blank and a series of standard solutions of 0, 1, 2, 3, and 5 mg/L were prepared from the 100 mg/L intermediate solution. Calibration standards for the analysis of Pb(II) in real water samples were prepared in 1% $HNO_3$.

*2.6. Batch Adsorption Studies*

The optimum adsorption conditions for SCB and OPS individually and in a homogenised combination for the removal of Pb(II) were determined by investigating the effects of pH, contact time, adsorbent dosage, and the initial metal ion concentration. The prepared simulated and real water samples were all stirred at 400 rpm at 25 °C. The pH of the samples was adjusted using 0.1 M and 0.5 M $HNO_3$ and 0.1 M and 0.5 M NaOH. The solutions were filtered two times using filter paper with a pore size of 0.45 μm. The residual metal ion concentration of the solution was determined using a FAAS. All batch adsorption studies were conducted in triplicate.

2.6.1. Removal of Pb(II) Using SCB and OPS Individually
Effect of pH

The effect of pH on 50 mg/L Pb(II) adsorption was investigated at pH 3, 4, 5, 6, and 7. The pH of each 100 mL solution was adjusted accordingly. Into each beaker, 0.05 g of SCB was added, and the solutions were stirred for 60 min. The solutions were filtered and analysed using FAAS. The same procedure was followed to remove 50 mg/L of a 100 mL Pb(II) solution using OPS as the adsorbent. The effect of pH could not be explored beyond pH 7 as precipitation was observed in alkaline solution.

Effect of Contact Time

The pH of 100 mL of a 50 mg/L Pb(II) solution was kept at 7 for the contact time study, and 0.05 g of SCB and OPS (individually) was added. The solutions were agitated for 10, 30, 60, 90, 120, 150, 180, and 210 min to investigate the effect of contact time. After the assigned contact time, the mixtures were filtered and analysed using FAAS.

Effect of Adsorbent Dosage

To evaluate the effect of adsorbent dosage, a 100 mL, 50 mg/L Pb(II) working solution was transferred in a 100 mL beaker, and the pH of the samples was adjusted to pH 7. An adsorbent amount of 0.01, 0.05, 0.1, 0.2, 0.25, 0.3, and 0.35 g of SCB was added into separate beakers, stirred at the optimum contact time of 60 min, filtered, and analysed by FAAS. The same procedure was followed to remove 50 mg/L of Pb(II) in a 100 mL solution using OPS. However, the effect of adsorbent dosage was investigated in the range of 0.01 to 0.23 g to preserve the aesthetic quality of water, and samples were stirred for 120 min. The solutions were then filtered and stored for analysis by FAAS. For subsequent studies, 0.2 and 0.17 g of SCB and OPS were used, respectively, to preserve the aesthetic quality of the water samples.

Effect of Initial Metal Ion Concentration

The influence of the initial metal ion concentration on the removal of Pb(II) at varying concentrations in 100 mL was carried out by preparing simulated water samples with initial Pb(II) concentrations in the range of 10 to 200 mg/L. The pH of each sample was adjusted to pH 7, and 0.2 g of SCB was added, stirred for 60 min, filtered, and analysed by FAAS. The same procedure was followed to remove 10 to 200 mg/L of Pb(II) in a 100 mL solution using 0.17 g of OPS and stirred for 120 min. After the appropriate contact time, the mixtures were filtered and analysed by FAAS to determine the residual Pb(II) concentration.

2.6.2. Removal of Pb(II) Using a Combination of Homogenised SCB and OPS Biosorbents

Effect of Contact Time

Here, 0.2 g of homogenised SCB and OPS (5:5) was used for the removal of 20 mg/L of Pb(II) in a 100 mL solution, at pH 7 at varying contact times. The effect of pH on 20 mg/L Pb(II) removal using homogenised biosorbents was not investigated as a result of optimal adsorption efficiencies at pH 7 for both adsorbents. Therefore, pH 7 was used throughout all Pb(II) sorption studies using a homogenised combination of SCB and OPS. The contact time was varied from 10 to 240 min. The optimum contact time was then determined by analysing the filtrate using FAAS.

Effect of Adsorbent Dosage Ratio Variation

The effect of using a combination of the adsorbents at varied ratios was investigated by varying the adsorbent dosage ratio for 0.2 g of SCB to OPS by ratios of 9:1, 8:2, 7:3, 6:4, 5:5, 4:6, 3:7, 2:8, and 1:9. These ratios were used to remove 20 mg/L of Pb(II) at pH 7, at the optimum contact time of 120 min. The optimum adsorbent dosage ratio variation was then determined by analysing the filtrate using FAAS.

Initial Metal Ion Concentration

The influence of the initial metal ion concentration was evaluated by preparing 100 mL of Pb(II) at different concentrations. The initial metal ion concentration ranged from 10 to 150 mg/L Pb(II) solutions. All Pb(II) solutions were adjusted to pH 7. An adsorbent dosage of 0.2 g of homogenised SCB and OPS at an optimum 5:5 adsorbent dosage ratio was added. This was followed by the agitation of adsorbent with Pb(II) solutions for the optimum contact time of 120 min. The solutions were then filtered and analysed to compare homogenised SCB and OPS removal efficiency at different concentrations.

2.6.3. Effect of Real Water Sample Matrix

To investigate the effect that the sample matrix has on the removal of Pb(II), 100 mL of real water samples from a borehole in Giyani, Limpopo province, were spiked with 10 mg/L of Pb(II). The pH of the solutions was adjusted as shown in Table 1. An adsorbent amount of 0.2 g of SCB, 0.17 g of OPS, and 0.2 g of a homogenised combination of SCB and OPS (5:5) was used. The mixtures were stirred for the contact time shown in Table 1. The mixtures were then filtered and stored for analysis by FAAS. Real water samples were acidified in 1% $HNO_3$ after sample collection. Therefore, all standards and blank solutions were prepared in 1% $HNO_3$.

**Table 1.** Optimum pH and contact time used for removal of Pb(II) in real water samples.

| Parameters | |
|---|---|
| pH | SCB: 7 |
| | OPS: 7 |
| | SCB:OPS (5:5): 7 |
| Contact time (min) | SCB: 60 |
| | OPS: 120 |
| | SCB:OPS (5:5): 120 |

### 2.6.4. Regeneration of SCB and OPS

For the regeneration of SCB and OPS, 10 mg/L of Pb(II) solutions were prepared, and the solution pH was adjusted as shown in Table 1. In the first cycle, the adsorbent dosages of 0.2 g for SCB, 0.17 g for OPS, and 0.2 g for homogenised SCB and OPS (5:5) were used for regeneration studies. The solutions were agitated to the contact time presented in Table 1, filtered, and analysed by FAAS. The adsorbents were recycled by stirring the remaining amounts of SCB, OPS, and the homogenised combination of SCB and OPS for an hour in 20 mL of 0.3 M of $HNO_3$. Significant amounts of sorbent material were lost after each cycle and all adsorption–desorption cycles were carried out in triplicate.

### 2.7. *Evaluation of Analytical Figures of Merit*

### 2.7.1. Determination of Limit of Detection, Limit of Quantification, and Linearity

The limit of detection (LOD) was determined by agitating 10 reagent blanks with SCB, OPS, and homogenised SCB and OPS under their respective optimum experimental conditions. The resulting filtrate absorbances was measured using FAAS. The LOD for each adsorbent is a product of 3 times the standard deviation ($3\sigma$) of the mean of the 10 reagent blank concentrations. The limit of quantification (LOQ) is calculated by multiplying the standard deviation of the average of reagent blank concentrations by 10 ($10\sigma$) [31–33]. The linearity was determined using the external calibration method. A series of Pb standards in the linear concentration range of 1 to 5 mg/L were prepared for the external calibration method. The coefficient of determination ($R^2$) and equation of the calibration curve were recorded and used to determine the acceptable linearity for the quantification of Pb(II) [32].

### 2.7.2. Determination of Method Accuracy and Precision

The assessment of method accuracy was determined at $1 \times$ LOQ and $10 \times$ LOQ levels. For $1 \times$ LOQ, real water samples of Pb(II) were spiked with 0.364 mg/L of Pb(II), whereas method accuracy at $10 \times$ LOQ was determined by spiking real water samples with 3.64 mg/L of Pb(II). The samples were analysed using FAAS and the percentage recovery was determined. All standards and blank solutions were prepared in 1% $HNO_3$. The method precision was evaluated by calculating the percentage relative standard deviation (%RSD) of samples run in triplicate.

## 3. Results and Discussion

### 3.1. *Characterisation of Biosorbents*

### 3.1.1. Zeta Potential Analysis

The zeta potential of SCB and OPS was studied in the pH range of 1 to 12, as presented in Figure 1. The zeta potential of adsorbent particles is determined using the potential difference in a capillary cell in which the sample is suspended to determine the velocity of suspended particles [34]. Zeta potential provides information on the charge of the adsorbent at a particular pH level. The point of zero charge (pzc), otherwise known as the isoelectric point, is the pH at which the material has a net charge of zero [35]. The pzc for SCB was found at pH 2.2, and the pzc for OPS was found at pH 1.27. Both SCB and OPS assume negative charges at any pH above the pzc [8].

The surface charge of an adsorbent can be altered by surface protonation and surface deprotonation. The protonation and deprotonation of the surface is influenced by the $H^+$ and $OH^-$ ions' concentration of the aquatic environment [8]. The electrical charge of the sorbent material becomes increasingly negative as the pH of the solution increases. The observed trend is the case with most adsorbents. The zeta potential of the solution becomes more negative with an increase in $OH^-$ concentration. The $OH^-$ ions make the surface of the adsorbents (stern layer) more negative by creating a negatively charged double layer [36]. Similarly, an acidic solution has a higher hydronium ($H^+$) ion concentration than $OH^-$ concentration; therefore, the double layer is positive [35–37]. It is also important to note that the surface of SCB and OPS assumes a negative charge in an aqueous solution due to ionisation of their acidic functional groups [36]. The zeta potential of sorbent material has

a significant influence on the sorption of ions. The positively charged ions will be attracted when the double layer of sorbent material bears a negative charge. Similarly, negatively charged ions will be attracted when the double layer has a net positive charge [35,37].

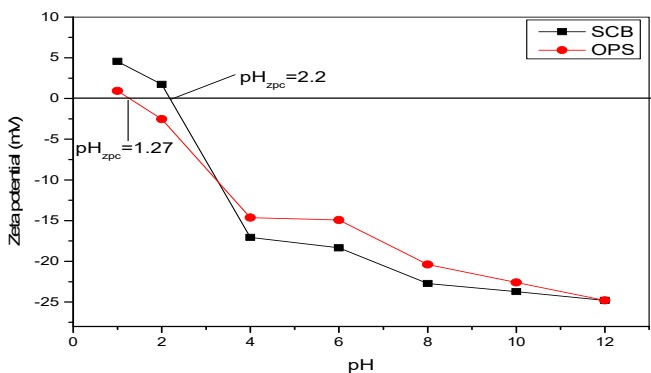

**Figure 1.** Zeta potential of SCB and OPS.

3.1.2. FTIR Spectroscopy Analysis

The FTIR spectra of SCB and OPS are compared in Figure 2. The use of FTIR as a characterisation technique allows the identification of key functional groups that could be responsible for adsorption. The FTIR spectra of SCB and OPS show high complexity, indicating a high number of functionalities. It is worth noting that the FTIR spectra of SCB and OPS typically have the same functionalities. The broad bands at 3327 cm$^{-1}$ for SCB and 3366 cm$^{-1}$ for OPS justify the presence of an alcohol (O-H) and a pectic acidic group [38–40]. The peak at approximately 1600 cm$^{-1}$ in both spectra corresponds to carbonyl (C=O) stretching. The carbonyl group and hydroxyl group are associated with carboxylic acids or phenolics. It is also important to note that bands at approximately 1600 cm$^{-1}$ may be assigned to C=C stretching found in graphene [24,40].

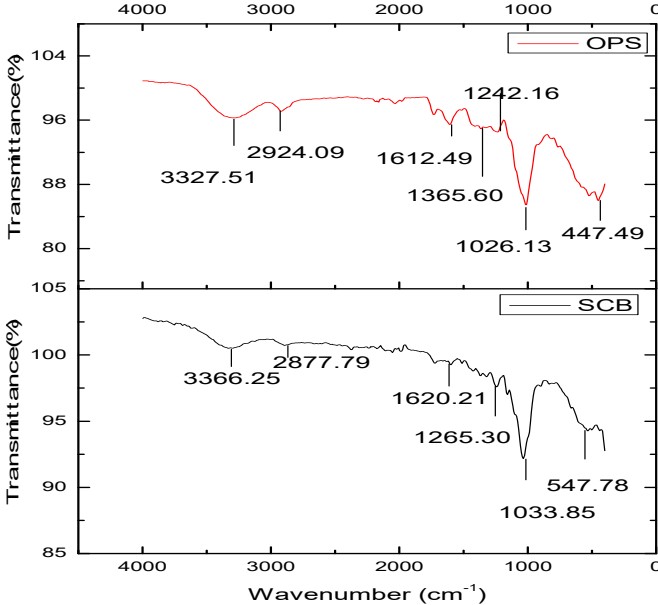

**Figure 2.** FTIR spectra of SCB and OPS.

It is also important to note that the bands at 2877 and 2924 cm$^{-1}$ are assigned to symmetric and asymmetric C-H2 stretching, generally belonging to aliphatic groups [5]. Bands found at 1242 and 1265 cm$^{-1}$, for SCB and OPS, respectively, correspond to C-C and C-O stretching, which are typical for alcohols, carboxylic acids, esters, and ethers [35,39]. Conspicuous absorption at approximately 1000 cm$^{-1}$ in both adsorbents is attributed to

Si-O stretching, which indicates the possibility of silica being present [41]. The bands at 1000 cm$^{-1}$ may also indicate C-C, C-OH, and C-O bands [39]. It can be justified that the main functional groups that facilitate adsorption of metals are oxygen-containing functional groups, and these include functionalities such as carbonyl, carboxyl, and hydroxyl groups [42]. Bands found at 600 cm$^{-1}$ correspond to metal-halogen (M-X) vibrations. When compared to FTIR spectra of several studies, the spectra indicate the presence of lignin, pectin and cellulose, and hemicellulose [15,40,43]. Both spectra could be compared to those reported in [39,40]. A study by Licona-Aguilar alluded that the intensity of the FTIR peaks can be correlated with the quantity of cellulose and hemicellulose. In this study, OPS demonstrated greater intensities for absorptions at 3327, 2924, and 1620 cm$^{-1}$ [43]. Lastly, using the FTIR library, both SCB and OPS spectra were matched with cellulose and hemicellulose spectra, which consist of functionalities that have a high affinity for metal ions [15].

### 3.1.3. TGA

The TGA was used to study SCB and OPS thermal degradation as the temperature increases from room temperature to 1000 °C. Figure 3 depicts the TGA curves for SCB and OPS at a constant heating rate. Both biosorbents displayed a similar pattern. Thermal degradation of biosorbents is affected by physicochemical properties such as functionality, surface texture, crystallinity, fibre size, and temperature [37].

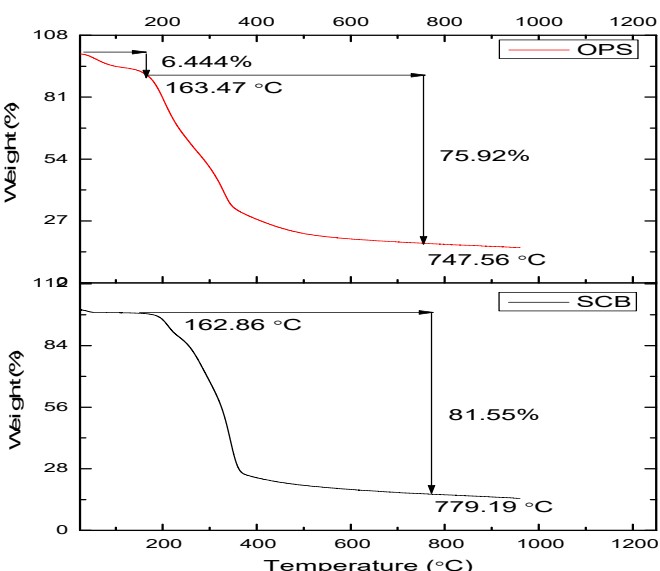

**Figure 3.** TGA thermograms of SCB and OPS.

The TGA curve makes it possible to determine the contents of ash [41]. Thermogravimetric analysis for OPS shows two mass-loss events. Approximately 6.44% of OPS was lost in the first event, and the second event had a significant weight loss of 75.92%. Between 25 and 163.47 °C, humidity and volatile matter were lost. Biomass decomposition occurred between 163.47 and 747.56 °C [40].

Unlike OPS, SCB had one weight-loss event. The event resulted in the loss of 81.55% from room temperature to 779.19 °C, and it is expected that humidity and biomass were lost in a single event [41]. The loss of biomass includes the structural components of hemicellulose, cellulose, pectin, and lignin [24,40]. Cellulose, hemicellulose, and pectin are the first to decompose. Succeeding the degradation of the first three components is lignin, where a significant portion of its composition could be phenolic compounds. Hemicellulose is lost at temperatures below 350 °C. Cellulose is lost throughout a heating range of 250 to 500 °C. At temperatures above 370 °C, the decomposition of lignin begins. Lignin continues to decompose at a constant weight-loss rate up to approximately 950 °C [24,40]. The loss of

these components happens concurrently; therefore, their degradation cannot be isolated into individual weight-loss events. The decomposition of biomass structural components occurring at temperatures above 163 °C was exothermic [40].

### 3.1.4. SEM and EDS Analyses

The SEM micrographs in Figures 4 and 5 were utilised to survey the morphology of the surfaces of SCB and OPS, respectively. The sorption of metals onto the surface of an adsorbent does not only rely on its chemical composition but also its morphological structure. The SCB microphotograph in Figure 5 reveals that SCB has an irregular structure and consists of sheets of narrow fibres with small pores that facilitate the adsorption of PTEs [35]. Sheets of narrow fibres and pores of different sizes and shapes were also observed in similar studies [5,21].

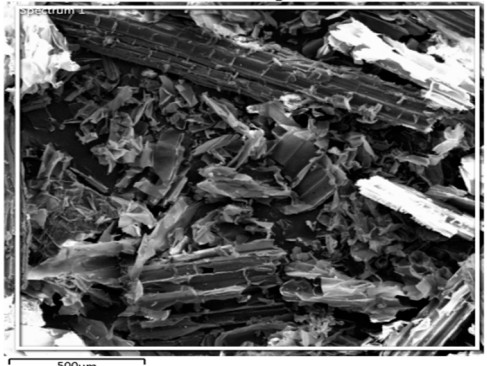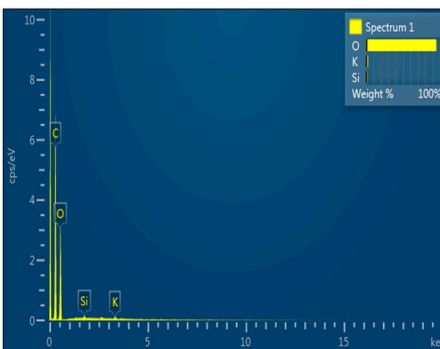

**Figure 4.** SEM images and EDS spectrum of SCB.

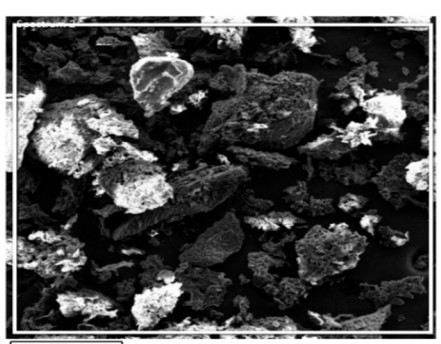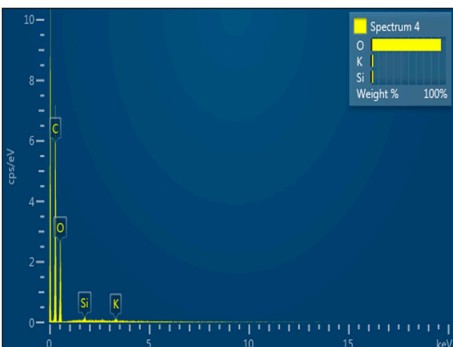

**Figure 5.** SEM images and EDS spectrum of OPS.

Figure 5 reveals that OPS has an extremely irregular-porous surface. The SEM image for OPS in Figure 5 demonstrated similar pores of different size and shape as observed by Kamsonlian et al. [41]. The inner white part of OPS, known as the albedo zone, is the reason OPS are highly porous [44]. Bhagat et al. reported that OPS are highly porous [14]. The EDS, an analytical technique, was used to gather information on the elemental composition of the biosorbents. Both adsorbents consist of O, Si, and K, where oxygen is dominant. The high O content in the EDS corresponds with the large number of high O-containing biomass structural components found in SCB and OPS. Structural components such as cellulose, hemicellulose, lignin, and pectin have a large number of oxygen atoms and compounds [41]. The presence of Si in the EDS also complements the Si-O stretching frequency identified in both spectra in FTIR (Figure 2). Lastly, the SEM micrographs of SCB and OPS revealed that both biosorbents are not spherical.

### 3.1.5. TEM Analysis

The application of TEM as a surface technique may be utilised to determine the particles' internal structure, arrangement, shape, length, diameter, and size [14,37,45]. This technique was used to complement results obtained in other characterisation techniques. However, the sorbent particles with different sizes and shapes interacting by weak intermolecular forces have affected the technique's ability to gather information on the sorbent material's size, shape, length, and diameter [46]. The only information that could be gathered from the TEM images (Figure 6) is that OPS are more porous than SCB, which is in agreement with SEM results.

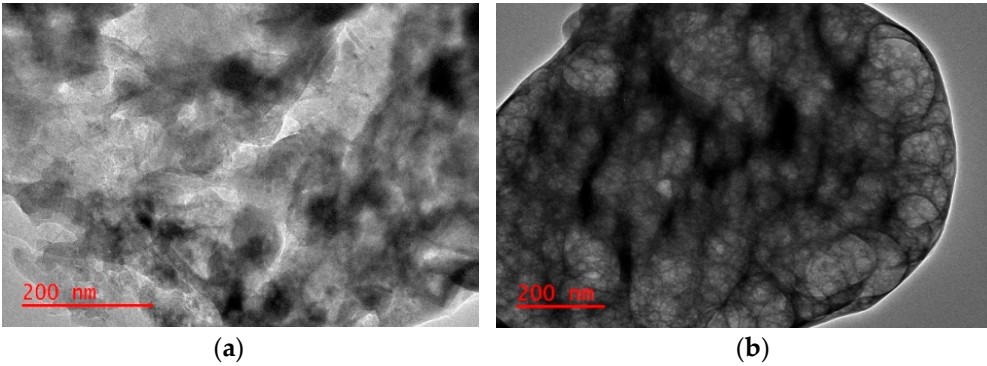

| (a) | (b) |

**Figure 6.** TEM images of: (**a**) SCB and (**b**) OPS.

### 3.1.6. pXRD Analysis

The crystallographic arrangement and chemical and physical composition of SCB and OPS were studied using pXRD. Figure 7 shows the pXRD diffractograms of SCB and OPS. The SCB and OPS pXRD diffractograms revealed a similar trend. A comparison of the SCB and OPS diffractograms with diffraction patterns in the pXRD database matched diffractograms of SCB and OPS to those of sucrose.

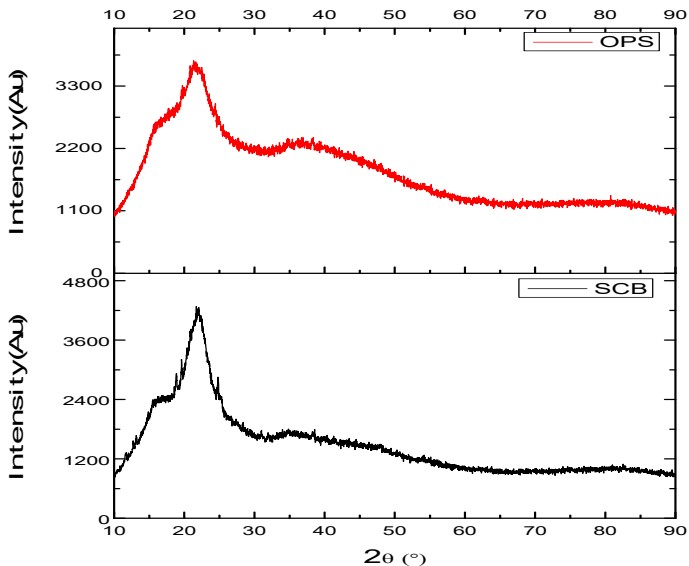

**Figure 7.** pXRD diffractograms of SCB and OPS.

The broad peaks clearly indicate that carbon in plant material makes the sorbent extremely amorphous [47]. The wide band found in the range of $2\theta = 15°$ and $2\theta = 28°$ is a clear indication of vitreous material, which is defined as solid, non-crystalline, and unstable matter [48]. A study by Licona-Aguilar et al. classified characteristic diffraction peaks between 15° and 20° to monoclinic β-polymorphous cellulose I [43]. The patterns exhibit

peaks that may be assigned to cellulosic non-crystalline plant material at a diffraction angle of $2\theta = 22.44°$ and $2\theta = 16.51°$ for SCB, and $2\theta = 22.44°$ and $2\theta = 16.13°$ for OPS. The sharp peaks at approximately $2\theta = 22°$ are typical for cellulose, and diffractogram peaks at around $2\theta = 16°$ may be assigned to hemicellulose [47].

### 3.1.7. BET Analysis

The surface area, pore size, and pore volume of SCB and OPS are presented in Table 2. The BET results indicate that SCB has a significantly greater surface area and pore size, while the pore volumes of OPS and SCB are relatively similar. Compared to other low-cost biosorbents such as neem leaves, rice husks, and oak bark, SCB and OPS have a small surface area [49–51]. Pore volume describes saturation of the pore with $N_2$ at the greatest relative pressure (P/P0), where P/P0 = 0.99 [52]. It is expected that SCB should adsorb more cations given that it has a larger surface area [39].

**Table 2.** BET measurements of SCB and OPS.

| Sample | Surface Area ($m^2/g$) | Pore Size (Å) | Pore Volume ($cm^3/g$) |
|---|---|---|---|
| SCB | 4.90 | 2417 | 0.30 |
| OPS | 1.24 | 131 | 0.22 |

### 3.2. Batch Adsorption Studies

### 3.2.1. Adsorption Studies Using SCB and OPS Individually, and in a Homogenised Combination of Bisorbents

### Evaluation of pH Effect on Adsorption

The effect of pH was investigated to provide a clear understanding of the role of pH in the adsorption of the selected toxic metal ions. The observed adsorption trend in Figure 8 revealed that the percentage removal and adsorption capacity of Pb(II) ions in solution increased with the increased sample pH for both adsorbents.

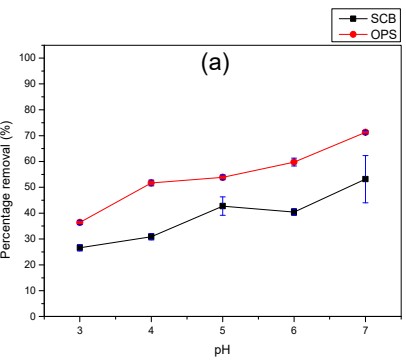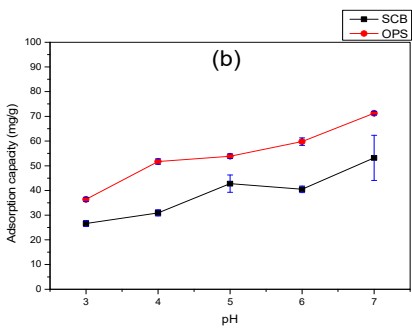

**Figure 8.** Effect of pH: (**a**) 50 mg/L Pb(II) percentage removal and (**b**) 50 mg/L Pb(II) adsorption capacity.

Adsorption of Pb(II) using OPS as an adsorbent showed a higher adsorption capacity and higher percentage removal levels as compared to adsorption using SCB. The pH study was concluded at pH 7 as a result of the observed precipitate at pH levels beyond pH 7. At pH 7, 53.19% and 71.45% removal levels were observed using SCB and OPS, respectively. The observed increase in adsorption from pH 3 to 7 is supported by Figure 2, which shows the zeta potential of SCB and OPS in the pH range of 1–12. The sorbent material becomes increasingly oxidised and consequently negative as the pH of the solution increases. As the pH of the contaminated simulated samples increases beyond the pzc of 2.2 for SCB and 1.27 for OPS, the adsorbents bear a negative charge and their affinity for positively charged species of Pb(II) increases.

As a result of the dominance of positively charged mononuclear Pb(II) ions and positively charged Pb(II) complex formation at any pH below 7.5, an increase in the affinity

of Pb(II) to the negatively charged surface of both SCB and OPS was observed due to electrostatic attraction. The sorption of positively charged Pb(II) increases as the pH increases due to the introduction of oxygen-bearing functionalities that act as Pb(II) ion anchors [53]. The low adsorption capacity of Pb(II) under acidic conditions is apparent as a result of the increased competition between H$^+$ ions in excess and positively charged Pb(II) ions for active binding sites [7,8,44]. The low Pb(II) adsorption under acidic conditions is due to the protonation of the adsorbents' surface, which may repel Pb(II) ions [7]. Several studies reported a similar trend; however, pH 6 was reported as the optimum pH for Pb(II) removal [13,14,39]. It is also worth noting that both adsorption capacity and percentage removal of Pb(II) increased with an increase in pH [44,54].

The homogenised SCB and OPS effect was investigated to evaluate and compare Pb(II) removal efficiencies using SCB and OPS individually and in a homogenised combination. Homogenised SCB and OPS may serve as a suitable alternative in the case that there is a shortage of any of the adsorbents due to seasonal variations. The pH study for the removal of Pb(II) using homogenised SCB and OPS was not conducted due to observations of maximum adsorption appearing at pH 7 for both adsorbents when the investigation was conducted individually. Hence, pH 7 was used for all subsequent studies for the removal of Pb(II) using a homogenised combination of SCB and OPS.

Evaluation of Contact Time Effect on Adsorption

It can be seen in Figure 9 that adsorption by SCB reaches a maximum capacity of 43.79 mg/g at a contact time of 60 min, where 43.79% of 50 mg/L of Pb(II) was removed. The OPS showed a maximal adsorption capacity of 78.08 mg/g, with up to 78.08% removal of 50 mg/L of Pb(II), at 120 min. After the maximum adsorption at 120 min, the percentage removal for Pb(II) using OPS decreased to 68.25% with an adsorption capacity of 68.25 mg/g at 150 min. The Pb(II) percentage removal and adsorption capacity remained relatively constant after 150 min. An adsorption took place at a fast rate within the first 120 min as a result of the availability of active sites on surfaces of the adsorbent, as reported by Badmus et al. [55]. As these active sites become saturated, the rate of adsorption decreases. The decrease in percentage removal of metal ions could also be attributed to the change of metal ion adsorption from ion-exchange to chemisorption [55]. It is expected that increments of contact time elicit a positive effect on the percentage removal of metal ions due to the saturation of biosorbent surfaces. However, a study by Maity et al. ascribed the decrease in removal of metal ions with an increase in contact time to a threshold contact time. The same study further explained that the observed effect can be described by a two-phase phenomena involving the rapid phase, which is later displaced by the delayed phase [56].

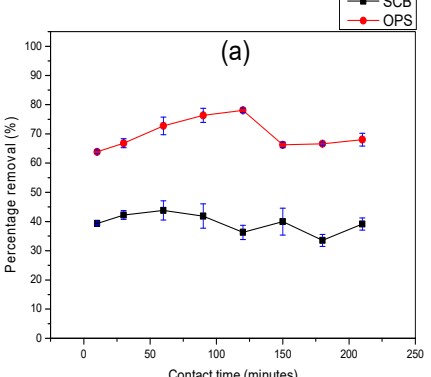 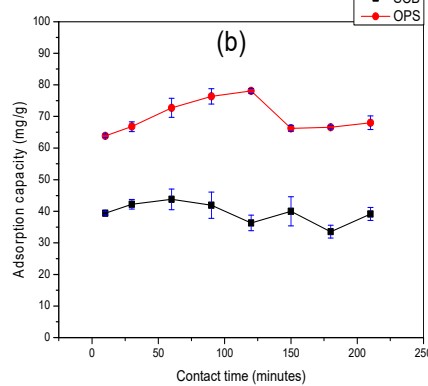

**Figure 9.** Effect of contact time: (**a**) 50 mg/L Pb(II) percentage removal and (**b**) 50 mg/L Pb(II) adsorption capacity.

The adsorption capacity and percentage removal for Pb(II) are high within the first 10 min as a result of the large concentration gradient and large accommodative surface area of SCB and OPS, as seen in BET results in Table 2 [57,58]. Studies have revealed that adsorption of metals increases with an increase in contact time [2,14,57]. These studies also revealed that saturation is often reached after 60 min [14,58].

In the contact time study for Pb(II) removal by a homogenised combination of SCB and OPS, an initial metal ions concentration of 20 mg/L was utilised as a result of the high Pb(II) percentage removal that was obtained when SCB and OPS were employed individually. Figure 10 depicts the influence of contact time for removing 20 mg/L of Pb(II) using a combination of homogenised SCB and OPS. The sorption of 20 mg/L of Pb(II) onto 0.2 g of SCB and OPS, at a 5:5 ratio, showed higher removal levels at 120 min. The percentage removal of 20 mg/L of Pb(II) was 98.08%. After 120 min, the percentage removal of Pb(II) showed a slight decrease to approximately 97% from 150 to 240 min.

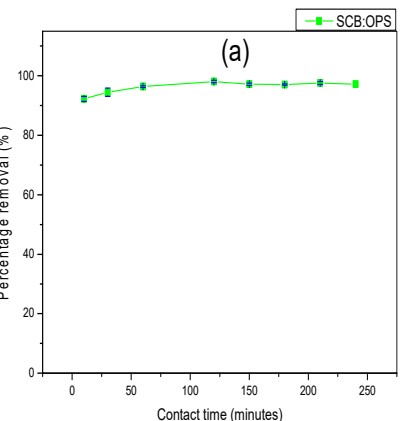 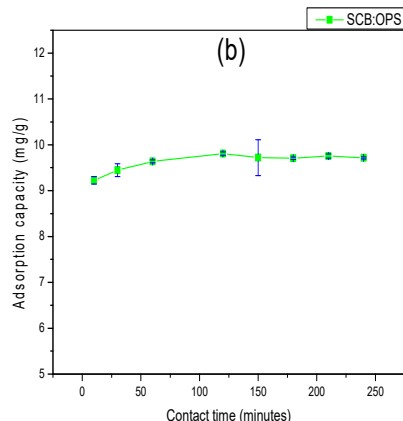

**Figure 10.** Effect of contact time: (**a**) 20 mg/L Pb(II) percentage removal and (**b**) 20 mg/L Pb(II) adsorption capacity.

The mixture of adsorbents showed maximum sorption capacity and 100% removal throughout the contact time range of 10 to 240 min. The relatively constant and high removal efficiency and adsorption capacity of Pb(II) by homogenised SCB and OPS is a clear indication that the sorbent material approaches complete saturation within 10 min. The observed rapid adsorption rate is attributed to the high surface area and affinity of biosorbents for Pb(II) ions in concentrated solutions. Once complete saturation is reached, the removal efficiencies of PTEs are determined by the rate at which PTEs ions are transported from the external to internal sites of the sorbent material [57].

Evaluation of Adsorbent Dosage Effect on Adsorption

The adsorbent dosage study revealed that the sorption of Pb(II) onto SCB and OPS increases at higher adsorbent dosages [14,58]. This study also revealed that adsorption capacity has an inversely proportional relationship to adsorbent dosage. The increase in Pb(II) adsorption is explained by the phenomenon that higher biosorbent dosages have a larger surface area and consequently, a larger number of active sites [2,14,15,39]. An increase in the number of active sites is indicative of an increase in the availability of oxygen-bearing functional groups such as alcohols, carboxylic acids, esters, and ethers that were identified in the FTIR spectra for both SCB and OPS. Figure 11 shows the relationship between adsorbent dosage and Pb(II). Figure 11 shows that 0.01 g of SCB and OPS had the highest adsorption capacity for Pb(II). However, the percentage removal efficiency of both biosorbents was the lowest at an adsorbent dosage of 0.01 g. Although high removal levels were observed at higher adsorbent dosages, only 0.2 and 0.17 g of SCB and OPS were used, respectively, for subsequent studies. This was carried out to ensure that the aesthetic quality of the water is preserved. Post-treatment, a colour change was observed at dosages higher than 0.2 g for SCB and 0.17 g for OPS.

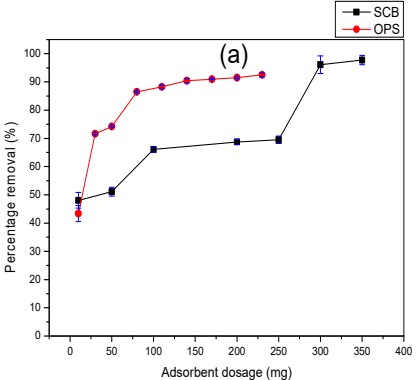 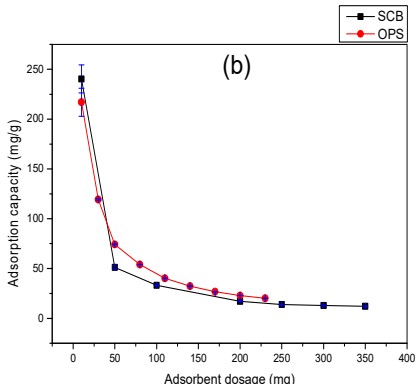

**Figure 11.** Effect of adsorbent dosage: (**a**) 50 mg/L Pb(II) percentage removal and (**b**) 50 mg/L Pb(II) adsorption capacity.

To remove 50 mg/L of Pb(II) using 0.2 g of SCB, 68.77% of Pb(II) was removed. On the other hand, the percentage removal of 50 mg/L of Pb(II) using 0.17 g of OPS was 91.00%. The maximum adsorption capacity using SCB and OPS was found at 0.01 g. The maximum adsorption capacity for Pb(II) removal using SCB and OPS was 240.36 and 216.99 mg/g, respectively.

Optimum percentage removal levels of 20 mg/L of Pb(II) ions have been reported using OPS at an adsorbent dosage of 0.8 g [14]. Although the study displayed different percentage removal efficiencies at different initial metal ion concentrations and optimum adsorbent dosages, the same trend of higher PTEs removal levels with the increased adsorbent dosage was observed.

The removal of 20 mg/L of Pb(II) was studied using 0.2 g of a homogenised combination of SCB and OPS. In the adsorbent dosage ratio study, the SCB and OPS were varied from a 9:1 to a 1:9 SCB to OPS ratio, as shown in Figure 12. The study revealed that sorption capacity and percentage removal of 20 mg/L of Pb(II) increased from a 1:9 to a 5:5 adsorbent dosage ratio. After a 5:5 SCB to OPS ratio, there was a decrease in the sorption capacity of Pb(II) from 9.70 to 9.53 mg/g at a 9:1 ratio. Therefore, the ideal homogenised combination of adsorbent dosage for removing 20 mg/L of Pb(II) is at a 5:5 (1:1) ratio, where the percentage removal was 97.01%.

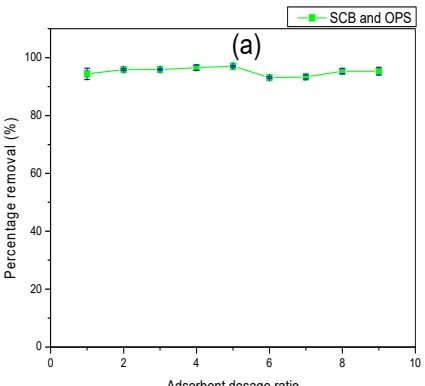 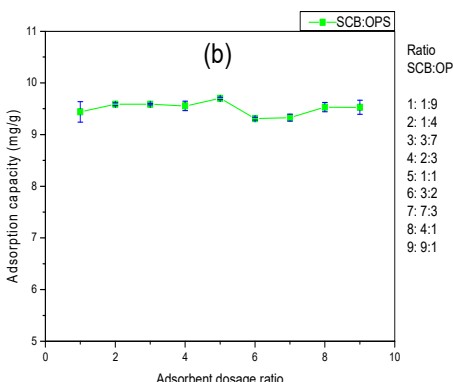

**Figure 12.** Effect of adsorption capacity: (**a**) 20 mg/L Pb(II) percentage removal and (**b**) 20 mg/L Pb(II) adsorption capacity.

The relatively constant removal efficiency of Pb(II) throughout the variation of SCB to OPS dosage ratio study is mainly due to SCB and OPS having similar structural components (cellulose, hemicellulose, and lignin) and functional groups [8,9,15,44]. The observed similarities are complemented by comparatively indistinguishable FTIR, pXRD, and EDS spectra for both SCB and OPS. The FTIR spectra revealed that both adsorbents had the same functional group and the EDS uncovered that both biosorbents had a concentration

of oxygen-bearing functional groups. The pXRD diffractograms disclosed that the SCB and OPS had a similar crystallographic arrangement, and thus, analogous physicochemical properties. For this reason, a variation in the adsorbent dosage ratio had no apparent influence on the adsorption of Pb(II) ions.

Evaluation of Initial Metal Ion Concentration on Adsorption

Figure 13 demonstrates the effect of the initial metal ion concentration on Pb(II) uptake. The percentage adsorption of Pb(II) decreases with an increase in the concentration of the selected toxic metal ions [14]. As the concentration of Pb(II) increases, the availability of active sites on biosorbents decreases. Although we were able to establish that SCB and OPS were highly porous and contained oxygen-rich structural components from the SEM-EDS analysis, the adsorption sites become saturated at higher initial metal ion concentrations due to an escalation in metal ions' competition for the available biosorbent active sites [57].

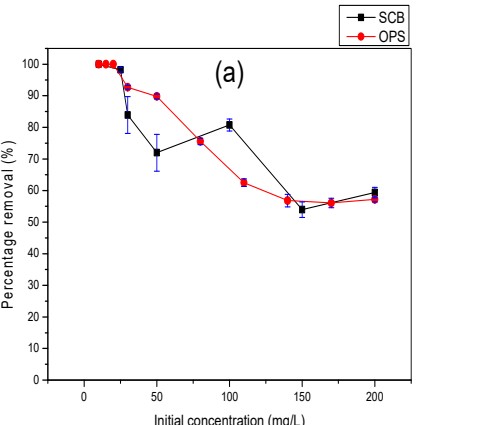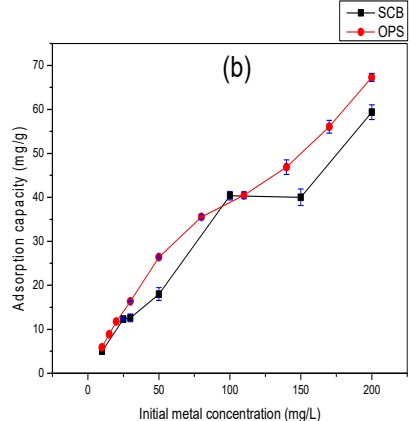

**Figure 13.** Effect of initial metal ion concentration: (**a**) Pb(II) percentage removal and (**b**) Pb(II) adsorption capacity.

Another possible reason that the adsorption of metal ions decreases at elevated initial metal ion concentrations is due to increments in the number of collisions between metal ions in solution. These collisions hinder the adsorption of metal ions onto the surface of sorbent material [2]. Using SCB, 100% removal levels were obtained for up to 10 mg/L of Pb(II). The OPS had the capacity to remove 100% of 20 mg/L of Pb(II). This is a clear indication that SCB and OPS have a large affinity for Pb(II) ions. The high adsorption efficiency of Pb(II) ions by both biosorbents is associated with the biosorbents' electronegativity in alkaline aqueous solutions. Under basic conditions, biosorbent surface functional groups are deprotonated and oxidised, thus resulting in negatively charged biosorbent surfaces. Hydrolysed Pb(II) species are predominately positive under basic conditions; therefore, there is an increase in the electrostatic interaction between cationic Pb(II) ions and negatively charged biosorbent surfaces [35–37].

Moreover, the high Pb(II) removal efficiencies were better elucidated by Licona-Aguilar et al., who identified Pb(II) ions in the form of metal hydroxides and carbonates using the metallic overlapping signals with XRD diffractograms of spent SCB and OPS [43]. The SEM images shown by Bharti and Kumar displayed a change in the morphology of spent SCB and OPS. As expected, the analysis of spent biosorbents by EDS tested positive for Pb(II)-loading [59]. In addition, the results showing the overlap between compounds of Pb(II) with pXRD diffractograms by Licona-Aguilar et al. serve as evidence that Pb(II) was successfully adsorbed by previously identified oxygen-bearing functionalities of cellulose and hemicellulose that were established through the analysis of FTIR, TGA, and EDS data in this study [43].

Figure 14 shows the initial metal ion concentration study for homogenised SCB and OPS. The adsorbent dosage ratio of 5:5 was used for the removal of Pb(II) ions at varying metal ion concentrations. The adsorption of Pb(II) was investigated between 10 and

150 mg/L initial metal ion concentrations. The Pb(II) adsorption percentage removal decreased from 100% for 10 mg/L to 63.78% for the removal of 150 mg/L. The adsorption capacity of the biosorbents increased as the initial metal ion concentration increased. This is attributed to an increase in the metal ions' interaction with the surface of the biosorbents. The decrease in percentage removal levels as the initial concentrations of Pb(II) increased is due to the heightened competition for binding sites on the surface of homogenised SCB and OPS [14,58].

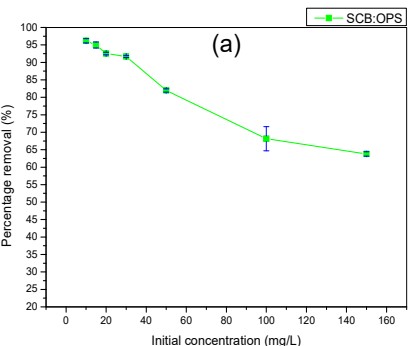 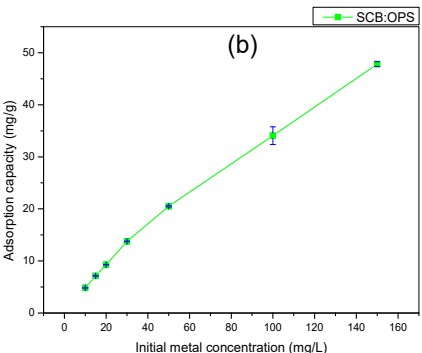

**Figure 14.** Effect of initial metal ion concentration: (**a**) Pb(II) percentage removal and (**b**) Pb(II) adsorption capacity.

### 3.2.2. Adsorption of Pb(II) in Real Water Samples

The remediation of water contaminated by Pb(II) in real water samples was investigated to study the effect of the sample matrix. The effect of the real water sample matrix was surveyed to assess the performance of the sorbent material in the presence of possible competitor ions. The real water samples were spiked with 10 mg/L of Pb(II) due to concentrations of Pb(II) in the borehole water samples being below the LOD of the detection technique. The borehole water from a village in Giyani contained anions such as nitrates ($NO_3^{3-}$), phosphates ($PO_4^{3-}$), sulphates ($SO_4^{2-}$), fluorides ($F^-$), and chlorides ($Cl^-$). The borehole water also contained cations such as Na(I), Ca(II), and Mg(II) in the concentration range of 20 to 241 mg/L [60].

The Pb(II) ions predominantly exist as cations in an aqueous solution [21]. Therefore, common ions that may exhibit a positive charge such as $Na^+$, $Mg^{2+}$, and $Ca^{2+}$ may compete with Pb(II) ions for adsorption at sorbent active sites. Under the optimum experimental conditions established using simulated water samples, the removal of 10 mg/L of Pb(II) in real water samples remained at 100% using SCB and OPS individually and in a homogenised combination, as demonstrated in Figure 15. The sample matrix of real water samples had no notable effect on the adsorption of 10 mg/L of Pb(II) under optimum experimental conditions. These observations demonstrate that SCB and OPS have a high affinity for PTEs such as Pb(II) in the presence of potential competitor ions.

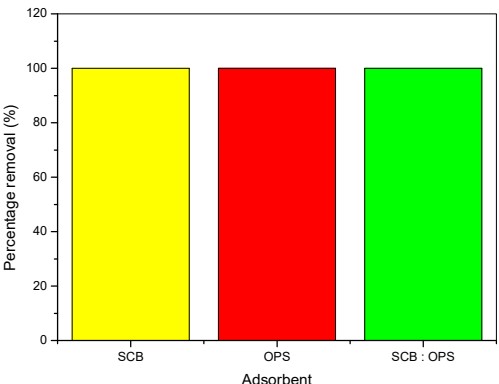

**Figure 15.** Effect of real water sample matrix on 10 mg/L Pb(II) percentage removal.

### 3.2.3. Comparison of Pb(II) Removal by Biosorbents

Table 3 shows the optimum experimental conditions for the remediation of water contaminated by Pb(II) using SCB and OPS individually and in a homogenous combination. Batch adsorption studies revealed that OPS had a greater removal efficiency for Pb(II) ions. The marginally greater vibration intensities in the FTIR spectra for OPS demonstrate that OPS have larger amounts of cellulose and hemicellulose which contain colossal concentrations of oxygen-bearing functionalities [43]. Therefore, it is for this reason that OPS promotes higher adsorption efficiencies for Pb(II) ions. A comparison of the results shown in Table 3 also displays that SCB, OPS, and homogenised SCB and OPS could achieve high removal efficiencies. A comparison of the adsorbents used individually and in a homogenised combination did not have a significant influence on the amount of Pb(II) adsorbed.

**Table 3.** Summary of optimum experimental conditions for Pb(II) removal using SCB and OPS individually, and a homogenised combination of SCB and OPS.

| Parameters | SCB | OPS | SCB:OPS |
|---|---|---|---|
| pH | 7 | 7 | 7 |
| Contact time (min) | 60 | 120 | 120 |
| Adsorbent dosage (g) | 0.2 | 0.17 | 0.2 (1:1) |
| Initial metal ion concentration for 100% removal (mg/L) | 10 | 20 | 10 |

The similarities in the amounts of Pb(II) adsorbed are by virtue of the analogous physicochemical properties of the adsorbents as they appear in zeta potential, FTIR, TGA, pXRD, and EDS characterisation techniques. However, it is worth noting that OPS had 100% removal levels for 20 mg/L of Pb(II), whereas homogenised SCB and OPS could only remove up to 92.6% of 20 mg/L of Pb(II). Lastly, 100% removal efficiencies for 10 mg/L of Pb(II) could be obtained in the presence of the sample matrix in real water samples.

### 3.2.4. Regeneration of SCB and OPS

Reusability is an essential aspect of the techniques, methods, and materials used in the remediation of water. Therefore, the reusability of SCB and OPS for the remediation of water contaminated by Pb(II) was investigated by performing several adsorption–desorption cycles. The efficiency of the regeneration of sorbent material is measured by the number of cycles that can be carried out by each adsorbent and the percentage removal of metal ions after each successive cycle [15].

Neutral to slightly alkali conditions were ideal for the adsorption of Pb(II) ions. Therefore, the desorption of the selected toxic metal ions was conducted under acidic conditions. The SCB could be used for four adsorption–desorption cycles and OPS could only be repeated for three. Similarly, homogenised SCB and OPS could be used for three cycles. Figure 16 shows Pb(II) percentage removal after each successive regeneration cycle.

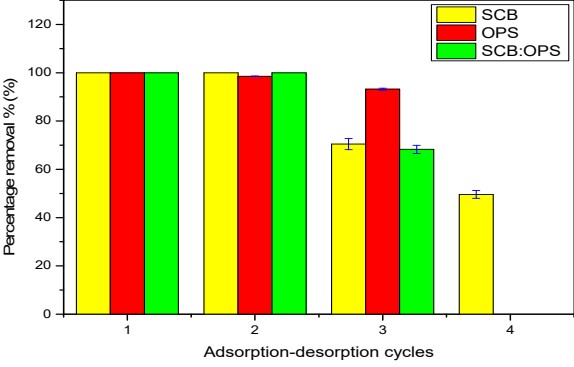

**Figure 16.** Pb(II) adsorption–desorption cycles.

It is crucial to acknowledge that the adsorbent mass decreases with an increase in the number of the regeneration cycles. The loss of biosorbent mass and consequent decrease in surface area and active sites may be attributed to the loss of biosorbent during filtration in batch adsorption studies, and filtration and drying after metal ions' desorption in 0.3 M $HNO_3$ [61]. Several studies have revealed that adsorption efficiencies of PTEs generally decrease after each adsorption–desorption cycle [21,62,63]. The use of strong acids such as $HNO_3$ enables the detachment of the Pb(II) ions from the biosorbents oxygen-bearing functional groups pre-determined in the FTIR, TGA, SEM-EDS, and pXRD analysis. The purpose of employing strong acids for the regeneration of the spent biosorbents is to ensure that the Pb(II) ions attached to adsorbent surfaces by weak interactive and electrostatic attraction (on oxygen-bearing functionalities) are weakened, which in turn enables successive biosorbent reusage [64].

The SCB had 100% removal efficiency for Pb(II) in the first two cycles. A decrease in percentage removal was observed in the third and fourth cycles. The SCB showed a percentage removal of Pb(II) ions of 70.47% in the third cycle and 49.63% in the fourth adsorption–desorption cycle. Using OPS, Pb(II) percentage removal levels were 100%, 98.57%, and 93.22% for the first, second, and third regeneration cycles, respectively. Although Pb(II) percentage removal levels were high throughout all adsorption–desorption cycles, the fourth cycle could not be conducted due to the increased loss in sorbent biomass. Homogenised SCB and OPS showed a percentage removal of 100% for the first two cycles and a percentage removal of 68.26% in the third cycle.

### 3.2.5. Comparison of Adsorption Capacities of SCB, OPS, and Homogenised SCB:OPS with Other Biosorbents

Table 4 shows agricultural biosorbents commonly applied for the remediation of water contaminated by Pb(II) in several studies, respectively. Table 4 includes the optimum experimental conditions, initial metal ion concentration, Pb(II) percentage removal levels, and attainable biosorbent adsorption capacities. The majority of the cellulose-based adsorbents indicate optimum Pb(II) adsorption at a pH greater than 5. This observation is elucidated by altering surface charge when hydroxide ions are in contact with the surface of adsorbents. The introduction of hydroxide ions in solution results in the deprotonation of surface functionalities [65]. An increase in the adsorbent dosage eventuates a larger surface area and consequent increase in active surface sites [14]. The studies have also argued that a higher initial metal ion concentration increases the competition for metal anchoring sites, and hence a decrease in adsorption efficiencies [57]. Lastly, the studies have displayed different optimum contact times ranging from 15 min to 24 h.

The reusability performance of cellulose-based agricultural adsorbents is compared in Table 5. The general trend is a decrease in adsorption efficiencies with an increase in adsorption–desorption/regeneration cycles. In some studies, the adsorption efficiency remained above 80% [39,66]. This could be related to the activation of sorbent surfaces after each regeneration cycle, where functional groups such as hydroxyl and carboxyl groups are highly active in the desorption of PTEs under acidic conditions [61,67].

The depletion of surface area and active sites is a possible explanation in the case where the adsorption efficiencies decreased after each adsorption–desorption cycle [67]. This indicates that a considerable amount of adsorbed metal ions on spent adsorbents is irreversible [61]. Chemical reagents such as acids and bases are used to treat spent adsorbents due to their potential to desorb metal ions on adsorbent surfaces. Chemicals such as NaOH, $HNO_3$, $H_2SO_4$, HCl, $NaNO_3$, and $H_3PO_4$ alter the pH of the solution and induce the solubilisation of adsorbed metals. The oxidation of adsorbent surfaces also enables the detachment of PTEs on spent adsorbents [68]. Most agricultural adsorbents were regenerated using acids such as HCl and $HNO_3$ due to their high regeneration efficiency [68].

**Table 4.** Comparison of adsorption of Pb(II) by various cellulose-based agricultural waste adsorbents under optimum experimental conditions.

| Biosorbent | Initial Concentration (mg/L) | pH | Contact Time (minutes) | Adsorbent Dosage | Percentage Removal (%) | Adsorption Capacity (mg/g) | Reference |
|---|---|---|---|---|---|---|---|
| Wheat bran | 200 | 7 | 60 | 10 g/L | 98.4 | 49.2 | [69] |
| Rice husks | 0.05 | 9 | 60 | 1 g/30 mL (33.33 g/L) | 96.8 | 0.0622 | [70] |
| Bark | 10 | 5 | 60 | 7.5 g/L | 86.7 | 88.5 | [71] |
| Banana peels | 50 | 5 | 20 | 40 g/L | 85.3 | 2.18 | [72] |
| OPS | 6 | 5 | 15 | 1 g/L | >40 | 27.9 | [39] |
| SCB | 100 | 6 | 120 | 10 g/L | 23.4 | 23.8 | [73] |
| SCB | 10 | 7 | 60 | 0.2 g/100 mL (2 g/L) | 100 | 5 | This study |
| OPS | 20 | 7 | 120 | 0.17 g/100 mL (1.7 g/L) | 100 | 11.8 | This study |
| Homogenised SCB and OPS | 10 | 7 | 120 | 0.2 g/100 mL (2 g/L) | 100 | 5 | This study |

**Table 5.** Comparison of removal efficiencies of Pb(II) using several regenerated cellulose-based agricultural adsorbents.

| Initial Pb(II) Concentration (mg/L) | Biosorbent | Dosage | Treated with | Number of Cycles | Adsorption Efficiency (%) | Reference |
|---|---|---|---|---|---|---|
| 50 | Fig sawdust | 0.5 g/50 mL (10 g/L) | 0.1 M HCl | 5 | 93 to 87 | [66] |
| 50 | SCB | 1 g/50 mL (1 g/L) | 0.1 M $HNO_3$ | 3 | 97 to 78 | [74] |
| 57 | SCB | 1 g/L | 1 M $HNO_3$ | 5 | 100 to >85 | [39] |
| 57 | OPS | 1 g/L | 1 M $HNO_3$ | 5 | 100 to >90 | [39] |
| 25 | Groundnut husk | 2 g/L | 0.1 M $H_2SO_4$ | 5 | 81.3 to 26.65 | [61] |
| 10 | SCB | 0.2 g/100 mL (2 g/L) | 0.3 $HNO_3$ | 4 | 100 to 49.63 | This study |
| 10 | OPS | 0.17 g/100 mL (1.7 g/L) | 0.3 $HNO_3$ | 3 | 100 to 93.22 | This study |
| 10 | Homogenised SCB and OPS | 0.2 g/100 mL (2 g/L) | 0.3 $HNO_3$ | 3 | 100 to 68.26 | This study |

3.2.6. Analytical Figures of Merit

Determination of Limit of Detection and Limit of Quantification

The LOD is the minimum detectable amount of analyte that can be determined by a method. The amount that can be detected with complete precision and accuracy is the LOQ [75]. The LOD and LOQ for the determination of Pb(II) in simulated water samples are recorded in Table 6. The LOD and LOQ were obtained using FAAS. The LOD and LOQ for Pb(II) determination by FAAS in the literature are reported in Table 7. The LOD and LOQ values recorded in Table 7 are significantly lower than the LOD and LOQ obtained for method validation in this study. The observed difference in LOD and LOQ could be attributed to differences in reagents' purity and instrumental settings [14].

**Table 6.** LOD and LOQ for Pb(II) determination.

| Adsorbent | LOD (mg/L) | LOQ (mg/L) |
|---|---|---|
| SCB | 0.109 | 0.364 |
| OPS | 0.287 | 0.957 |
| Homogenised SCB and OPS | 0.237 | 0.788 |

**Table 7.** LOD and LOQ of Pb(II) by FAAS in the literature.

| LOD | LOQ | Reference |
|---|---|---|
| 2.3 µg/L | 3.3 µg/L | [76] |
| 0.054 µg/L | 0.182 µg/L | [77] |
| 0.054 mg/L | 0.181 mg/L | [78] |

Linearity

The external calibration method was used to determine the linearity of the method. The $R^2$ of a plot enables comprehensive extrapolation and statistical analysis of analytes of unknown concentration. Data are considered acceptable and accurate when the $R^2$ of the plot is equal to 1 or close to 1. An $R^2$ of 1 indicates that the concentration of the analyte can be determined by extrapolation with 100% prediction [32].

This study tested the linearity for Pb(II) quantification in the linear concentration range of 1 to 5 mg/L. The $R^2$ of all Pb(II) analyses ranged between 0.9992 and 1.0000. This implies that unknown Pb(II) concentrations were extrapolated on the calibration curve with a prediction between 99.92% and 100% [32].

Evaluation of Method Accuracy

To evaluate method accuracy, real water samples were spiked with 0.364 mg/L of Pb(II) for 1 × LOQ and 1.14 mg/L for 10 × LOQ for Pb(II). The accomplished percentage recovery of Pb(II) at 1 × LOQ was 95.35%. At 10 × LOQ, the obtained percentage recovery of Pb(II) was 91.16%. According to the United States Environmental Protection Agency (US EPA), the recommended percentage recovery for method development and validation is between 75% and 125% [79]. Therefore, the percentage recovery value for Pb(II) complies with the regulatory US EPA certified guidelines for method development and validation [31]. The obtained percentage recovery is an indication that methods developed for the determination of Pb(II) are accurate. Furthermore, the percentage recoveries indicate that the method accuracy is within the acceptable range for determining Pb(II) ions in simulated and real water samples [33].

Evaluation of Method Precision

The percentage relative standard deviation (%RSD) of triplicate analysis for batch adsorption studies is used to determine the precision and repeatability of a method. The lower the %RSD, the higher the precision and repeatability of the method. A %RSD less than 15% is recommended for analysing analytes in replication [31–33]. The precision for the analysis of Pb(II) in batch adsorption studies was found in the acceptable range of 0.788% and 14.3%.

**4. Conclusions and Recommendations**

The SCB and OPS were successfully characterised using several characterisation techniques. Valuable information was drawn from zeta potential, FTIR spectroscopy, TGA, SEM-EDS, TEM, pXRD, and BET. The potential of SCB and OPS individually and in a homogenous combination was successfully evaluated and compared for the removal of Pb(II) ions in simulated and real water samples. The study revealed that Pb(II) uptake is highly pH-dependent. Both adsorbents showed a gradual increase in adsorption efficiencies with increased contact time. The uptake of Pb(II) ions increased with increments in adsorbent dosage. It was observed that under optimum experimental conditions, SCB and OPS showed 100% removal efficiencies for 10 and 20 mg/L of Pb(II), respectively. Therefore, it can be concluded that higher concentrations of Pb(II) can be removed under optimum experimental conditions using OPS. The homogenised combination of SCB and OPS revealed that 100% removal efficiencies for 10 mg/L of Pb(II) were attained at a 5:5 adsorbent dosage ratio under optimum experimental conditions.

The performance of SCB and OPS in isolation and in a homogenous combination remained at 100% for the removal of 10 mg/L of Pb(II) in the sample matrix of real water samples. The effect of multiple cations and anions in real water did not demonstrate any noticeable effect on the removal of 10 mg/L of Pb(II). The reusability performance of biosorbents individually and in a homogenous mixture demonstrated above 70% removal of Pb(II) ions from aqueous solutions for three adsorption–desorption cycles. This limited reusability up to three adsorption–desorption cycles could be attributed to the use of raw biosorbents. Furthermore, a change of colour of the water at a higher adsorbent dosage was observed in this study. For this effect, we recommend the use of activated forms of individual SCB and OPS as well as their combination in future studies. The method for removal of Pb(II) ions in water using biosorbents is simple, cheap, and scalable. If utilised on a large-scale basis, such a technique will create a fitting ecological environment for both humans and animals and create more employment amongst sugarcane and orange farms. This project will also substantially contribute to addressing issues pertaining to the disposal of waste products. This study promotes circular economy by using agricultural wastes as adsorbents for treatment of contaminated water and its reuse.

This study primarily focused on adsorption of Pb(II) from simulated aqueous solutions and Pb(II)-spiked borehole water using SCB and OPS individually and in a homogenous combination. To understand the underlying adsorption behaviour and its mechanism, investigations which entail adsorption isotherm and kinetic models are recommended in future studies.

**Author Contributions:** Conceptualisation, N.R.M. and A.A.A.; methodology, N.R.M. and A.A.A.; validation, N.R.M. and A.A.A.; formal analysis, N.R.M.; investigation, N.R.M. and A.A.A.; resources, A.A.A.; data curation, N.R.M.; writing—original draft preparation, N.R.M.; writing—review and editing, N.R.M. and A.A.A.; visualisation, N.R.M. and A.A.A.; supervision, A.A.A.; project administration, A.A.A.; funding acquisition, A.A.A. All authors have read and agreed to the published version of the manuscript.

**Funding:** Ntsieni Romani Molaudzi received financial support from the National Research Foundation (NRF), Grant Number MND19071845718457092. The project is supported in part by the NRF of South Africa under the Thuthuka Programme, Grant Number 117673, and the Water Research Commission (WRC) of South Africa, Project Number C2022/2023-00933.

**Data Availability Statement:** All the data used in this study have been reported in the manuscript.

**Acknowledgments:** The authors would like to acknowledge the University of Johannesburg Research Centre for Synthesis and Catalysis and Spectrum for the facility.

**Conflicts of Interest:** The authors declare no conflict of interest.

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
