# Peer review of "Sugarcane Bagasse and Orange Peels as Low-Cost Biosorbents for the Removal of Lead Ions from Contaminated Water Samples"

_water, doi:10.3390/w14213395_

Round 1

Reviewer 1 Report

The concept of using biomass waste as an adsorbent seems useful, and the manuscript is mostly well-organized. However, for possible improvement, the following comments are provided, please:

·         Many papers―rather more detailed and nuanced―dealing with the adsorption of lead onto either sugarcane bagasse or orange peels are already available in the literature (doi of some of the published papers are provided below as a few examples), and thus the novelty of the current manuscript is unclear. Although the authors are of the view that a combined usage of these two different adsorbents is a novelty feature (as mentioned in the introduction section), the reviewer thinks that this does not suffice as this does not add much value to the existing body of knowledge.

o   https://doi.org/10.3390/su141710860

o   https://doi.org/10.1016/j.bej.2006.07.001

o   https://doi.org/10.1016/j.sciaf.2021.e00931

o   https://doi.org/10.1016/S1003-6326(11)61309-5

o   https://doi.org/10.1016/j.solidstatesciences.2011.11.029

o   https://doi.org/10.1016/j.bej.2006.07.001

o   https://doi.org/10.2166/wst.2010.291

o   https://doi.org/10.1016/j.desal.2010.02.015

o   https://doi.org/10.1016/j.biortech.2015.06.006

·         Usually, the biomass is converted into activated carbon through some means, and then this activated carbon is used as the adsorbent, however, the authors have directly used the biomass as adsorbent, without converting it into activated carbon. The consequence is that the reusability of the adsorbent is low, and also the treated water becomes colored at higher dosage, indicating that the adsorbent is not very durable, and its disintegration, fragility, and dissolution could lead to secondary pollution (not just the release of dissolved adsorbent but also the release of desorbed pollutant into the treated stream) especially during consecutive adsorption-desorption cycles. In view of this, how would the authors justify the direct usage of biomass without any activation, and how about the aspect of secondary pollution due to dissolved adsorbent and the desorbed pollutant?

·         In adsorption-related studies, it’s mostly a typical approach to report the fitted adsorption isotherms (e.g., Langmuir or Freundlich) and the adsorption kinetics (e.g., pseudo first or second order model) which provides clues to the underlying adsorption behavior and the mechanism. However, no such studies have been reported in the manuscript. Inclusion of these is suggested.

·         The study is on the removal of lead, however, the introduction section is rather very general, and instead of discussing lead in particular, it discusses potential toxic elements (PTEs) in a very general manner. It is suggested to mention the pertinent aspects of lead, e.g., its occurrence (especially in the study area/country), exposure, and impacts.

·         The application area of this study is unclear: it is unclear whether the adsorbent is designed for surface water, groundwater, wastewater or industrial effluent. Because for the case of real samples, groundwater was used. But it is also mentioned that groundwater did not contain lead, and hence it was artificially spiked. But if this pollutant is not a problem in groundwater, then why such a study was required in the first place? How is this study environmentally-relevant and realistically-relevant? This needs to be duly justified.

·         Line 24: Instead of using the term ‘sustainability’, 'reusability' would be a more appropriate term, because sustainability is a wider concept which has not been thoroughly investigated in this study.

·         Line 141: Seems like a typo: it is unclear what is the pore size?

·         Line 453: “OPS being more porous than SCB”. This is not supported by the results in Table 2, which shows that SCB has a rather higher surface area, pore size, and pore volume. This contradiction needs to be corrected.

·         Figure 14b: The title of the x-axis seems wrong, and if so, it needs to be corrected.

·         QA/QC aspects, e.g., LoD, LoQ, linearity, etc. are supposedly part of the methodology section. It is unclear why these aspects have been included as a separation section in the Results and Discussion section.

·         Throughout the manuscript, symbol of dash (-) has been unnecessarily and wrongly used in words, e.g., po-table (line 51), en-ergy (line 69), reme-diation (line 70), pro-duce (line 75), or-ganic (line 77), and so on. This needs to be corrected thoroughly throughout the manuscript.

Author Response

 Reviewer 1

The concept of using biomass waste as an adsorbent seems useful, and the manuscript is mostly well-organized. However, for possible improvement, the following comments are provided, please:

  • Many papers―rather more detailed and nuanced―dealing with the adsorption of lead onto either sugarcane bagasse or orange peels are already available in the literature (doi of some of the published papers are provided below as a few examples), and thus the novelty of the current manuscript is unclear. Although the authors are of the view that a combined usage of these two different adsorbents is a novelty feature (as mentioned in the introduction section), the reviewer thinks that this does not suffice as this does not add much value to the existing body of knowledge.

o https://doi.org/10.3390/su141710860

o   https://doi.org/10.1016/j.bej.2006.07.001

o   https://doi.org/10.1016/j.sciaf.2021.e00931

o   https://doi.org/10.1016/S1003-6326(11)61309-5

o   https://doi.org/10.1016/j.solidstatesciences.2011.11.029

o   https://doi.org/10.1016/j.bej.2006.07.001

o   https://doi.org/10.2166/wst.2010.291

o   https://doi.org/10.1016/j.desal.2010.02.015

o   https://doi.org/10.1016/j.biortech.2015.06.006

Response: As indicated by the reviewer adsorption studies for removal of lead ions from aqueous solutions were reported, particularly using orange peels or sugarcane bagasse individually. In current study, we carried out characterisation of these biosorbents using a wide range of characterisation techniques and conducted adsorption studies using each of these biosorbents individually and homogenised combination of them. We believe investigation of a combination of two biosorbents is necessary since it provides alternative method for removal of lead ions from water when there is scarcity of biosorbents depending on seasons.  

  • Usually, the biomass is converted into activated carbon through some means, and then this activated carbon is used as the adsorbent, however, the authors have directly used the biomass as adsorbent, without converting it into activated carbon. The consequence is that the reusability of the adsorbent is low, and also the treated water becomes colored at higher dosage, indicating that the adsorbent is not very durable, and its disintegration, fragility, and dissolution could lead to secondary pollution (not just the release of dissolved adsorbent but also the release of desorbed pollutant into the treated stream) especially during consecutive adsorption-desorption cycles. In view of this, how would the authors justify the direct usage of biomass without any activation, and how about the aspect of secondary pollution due to dissolved adsorbent and the desorbed pollutant?

Response: In this study, adsorption capacity of raw biosorbents were investigated. We observed some limitations such as reusability to limited number of cycles and change of colour of water at a higher dosage. To this effect, we recommend the use of activated form of individual as well as composite of the orange peels and sugarcane bagasse. The conclusion section has been revised by pointing out these limitations and incorporating reviewer’s comments.

  • In adsorption-related studies, it’s mostly a typical approach to report the fitted adsorption isotherms (e.g., Langmuir or Freundlich) and the adsorption kinetics (e.g., pseudo first or second order model) which provides clues to the underlying adsorption behavior and the mechanism. However, no such studies have been reported in the manuscript. Inclusion of these is suggested.

Response: Studies focusing on adsorption isotherms and adsorption kinetics were not carried out in this study. Recommendations have been included as future studies in conclusions section.

  • The study is on the removal of lead, however, the introduction section is rather very general, and instead of discussing lead in particular, it discusses potential toxic elements (PTEs) in a very general manner. It is suggested to mention the pertinent aspects of lead, e.g., its occurrence (especially in the study area/country), exposure, and impacts.

Response: Introduction section has been revised as suggested by reviewer.

  • The application area of this study is unclear: it is unclear whether the adsorbent is designed for surface water, groundwater, wastewater or industrial effluent. Because for the case of real samples, groundwater was used. But it is also mentioned that groundwater did not contain lead, and hence it was artificially spiked. But if this pollutant is not a problem in groundwater, then why such a study was required in the first place? How is this study environmentally-relevant and realistically-relevant? This needs to be duly justified.

Response: The concentration of lead in the borehole water was below the limit of detection (LOD) of the method. The concentration of lead in borehole water was not detected by flame atomic absorption spectrometry due to limited sensitivity of the technique. As presented in section 3.2.6, the LODs ranged from 0.109 to 0.287 mg/L, which are far above the maximum permissible level (MPL) of lead in drinking water (0.01 mg/L). If more sensitive technique was used for borehole water analysis it could be detected and could be higher than the MPL and requires removal of the metal ion from water. In addition, borehole water sample spiked with lead was analysed to investigate the effect of the sample matrix. Therefore, spiking the borehole water samples enabled the assessment of the performance of the sorbent materials in the presence of competitor ions such as Na(I), Ca(II), and Mg(II) as presented in section 3.2.2. Furthermore, there is possibility of this method which could be extended for treatment of contaminated water from different sources.

  • Line 24: Instead of using the term ‘sustainability’, 'reusability' would be a more appropriate term, because sustainability is a wider concept which has not been thoroughly investigated in this study.

Response: The corrections have been conducted as recommended by reviewer.

  • Line 141: Seems like a typo: it is unclear what is the pore size?

Response: The typo has been rectified.

  • Line 453: “OPS being more porous than SCB”. This is not supported by the results in Table 2, which shows that SCB has a rather higher surface area, pore size, and pore volume. This contradiction needs to be corrected.

Response: The contradiction has been rectified.

  • Figure 14b: The title of the x-axis seems wrong, and if so, it needs to be corrected.

Response: Corrections have been conducted as suggested by reviewer.

  • QA/QC aspects, e.g., LoD, LoQ, linearity, etc. are supposedly part of the methodology section. It is unclear why these aspects have been included as a separation section in the Results and Discussion section.

Response: Methodology focuses on analytical procedures followed. The analytical procedures applied for evaluation of analytical figures of merit were detailed in section 2.7. Every experiment conducted following an analytical procedure has results. These results along with discussion were presented in section in section 3.2.6. Furthermore, evaluating analytical figures of merit such as LOD, LOQ, linearity, accuracy and precision is necessary in analytical method development.

  • Throughout the manuscript, symbol of dash (-) has been unnecessarily and wrongly used in words, e.g., po- table (line 51), en-ergy (line 69), reme-diation (line 70), pro- duce (line 75), or-ganic (line 77), and so on. This needs to be corrected thoroughly throughout the manuscript.

Response: All words unnecessarily divided by the dash have been corrected throughout the manuscript.

Reviewer 2 Report

In general this study is extensive regarding the characterization of the materials as well as the tests done on synthetic and real media and the reutilization of the materials. It is a complete study and it is worth publishing. However, it could be improved further.

1) The paper is well written. Some words are divided with a dash (-) for no obvious reason. Moreover, I am not sure of the choice of some terms like “foreign particles” (line 139) or “in isolation” (line 201)

2) The objective of the study as described at the end of the introduction is not attractive, since there are numerous studies with the same objective but with a different biomass. Therefore the novelty should be better described and convince the readers to continue with reading.

3) The units of the particle size (line 141) are not visible

4) The description of the various adsorbent  experiments should be more concise to avoid repetition. The use of tables to include all these information is a preferred way. Please consider to present in columns all parameters such as adsorbent (kind and dose), Pb concentration, contact time, pH etc. The different test conditions could be referred in each line. If one series of experiments aims at studying the effect of time, there could be a single line in the table with the range of contact times examined under the column of “contact time”.

5) Why is the removal of Pb decreasing after 120 min in the case of OPS? An explanation should be given. The conclusion about the when the maximum or steady values appear should be made after a statistical analysis which would prove that these points are actually higher than the adjacent ones (in the case of the maximum) or the same (in the case of the steady state). It seems that the maximum at 60 min is not obvious if we take the standard deviation of all the adjacent points. This analysis should be done with every graph which is used to make some conclusions about the maximum, minimum or steady values.

6) Try to combine the graphs for removal and adsorption capacity into one, showing these two different parameters at the left and right vertical axis separately (so that you can use different scale for the different range of values). In this way the visual correlation between these two parameters would be feasible.

7) I would suggest that the presentation for each factor (contact time, initial concentration) be in the same subsection for all tests; for the individual and combined adsorbents.  As it is now, the discussion about the differences is not facilitated since the reader has to go up and down the paper. The effect of the ratio of the two indiviadual materials to make the combined one could be presented in the subsection about the contact time to justify the selection of the ratio for the combined material. The presentation does not have to follow the sequence of the experiments as they have been performed.

8) Moreover, the discussion about the differences should be more distinct.

9) Please explain why not performing the tests for the effect of pH, dosage in the case of the combined adsorbent.

10) I think that the characterization tests (Figures 1-7) have not been used enough to explain the results shown in the adsorption and reutilization tests. They have been used to explain only the effect of pH in the case of the individual adsorbent and the initial concentration in the case of the combined adsorbent.

11) I suggest that you could also use these extensive characterisation tests to make comparisons between your results and the literature in the cases of studies which also involve a good characterisation of their materials as you did. With this comment, I would like to urge you to make the most of your results and enhance the merit of your paper, otherwise, the characterisation tests are useless.

Author Response

Reviewer 2

In general this study is extensive regarding the characterization of the materials as well as the tests done on synthetic and real media and the reutilization of the materials. It is a complete study and it is worth publishing. However, it could be improved further.

  • The paper is well written. Some words are divided with a dash (-) for no obvious reason. Moreover, I am not sure of the choice of some terms like “foreign particles” (line 139) or “in isolation” (line 201)

Response: The corrections have been made as suggested by reviewer.

  • All words unnecessarily divided by the dash have been corrected throughout the manuscript.
  • The phrase “foreign particles” has been replaced by “dust particles”.
  • The term “isolation” has been replaced by “individually” in the whole manuscript.

  • The objective of the study as described at the end of the introduction is not attractive, since there are numerous studies with the same objective but with a different biomass. Therefore the novelty should be better described and convince the readers to continue with reading.

Response: We tried to highlight the novelty of our study in the last paragraph of introduction section.

  • The units of the particle size (line 141) are not visible

Response:  The correction has been made.

  • The description of the various adsorbent experiments should be more concise to avoid repetition. The use of tables to include all these information is a preferred way. Please consider to present in columns all parameters such as adsorbent (kind and dose), Pb concentration, contact time, pH etc. The different test conditions could be referred in each line. If one series of experiments aims at studying the effect of time, there could be a single line in the table with the range of contact times examined under the column of “contact time”.

Response: We presented detail procedures instead of presenting optimisation parameters in ranges for anyone who interested to apply the method to follow the procedures easily.  

  • Why is the removal of Pb decreasing after 120 min in the case of OPS? An explanation should be given. The conclusion about the when the maximum or steady values appear should be made after a statistical analysis which would prove that these points are actually higher than the adjacent ones (in the case of the maximum) or the same (in the case of the steady state). It seems that the maximum at 60 min is not obvious if we take the standard deviation of all the adjacent points. This analysis should be done with every graph which is used to make some conclusions about the maximum, minimum or steady values.

Response: An explanation for the decrease after 120 minutes for OPS has been added in section 3.2.1.2. The results in figures have been presented by calculating standard deviations as indicated by error bars at each point.

  • Try to combine the graphs for removal and adsorption capacity into one, showing these two different parameters at the left and right vertical axis separately (so that you can use different scale for the different range of values). In this way the visual correlation between these two parameters would be feasible.

Response: The units and values on both the x-axis and y-axis for adsorption capacity (mg/g) and percentage removal (%) are different for some of the parameters. For example in Figure 10, the adsorption capacity is scaled up to 250 mg/g. Similar differences are also observed in Figures 12 and 13.  We fear that combining these graphs may be confusing to readers.

  • I would suggest that the presentation for each factor (contact time, initial concentration) be in the same subsection for all tests; for the individual and combined adsorbents. As it is now, the discussion about the differences is not facilitated since the reader has to go up and down the paper. The effect of the ratio of the two indiviadual materials to make the combined one could be presented in the subsection about the contact time to justify the selection of the ratio for the combined material. The presentation does not have to follow the sequence of the experiments as they have been performed.

Response: The sections have been combined as recommended by reviewer.

  • Moreover, the discussion about the differences should be more distinct.

Response: The characterisation techniques showed that SCB and OPS had similar physicochemical properties, however, we have expanded on the minor differences in several sections in the manuscript. We have also included a more descriptive explanation of the biosorbents similarities.

  • Please explain why not performing the tests for the effect of pH, dosage in the case of the combined adsorbent.

Response: Explanation for not conducting the pH study for a combined adsorbent has been included in section 3.2.1.1.

  • I think that the characterization tests (Figures 1-7) have not been used enough to explain the results shown in the adsorption and reutilization tests. They have been used to explain only the effect of pH in the case of the individual adsorbent and the initial concentration in the case of the combined adsorbent.

Response: The explanation using characterisation results have been used in adsorption and reutilisation studies of other sections.

  • I suggest that you could also use these extensive characterisation tests to make comparisons between your results and the literature in the cases of studies which also involve a good characterisation of their materials as you did. With this comment, I would like to urge you to make the most of your results and enhance the merit of your paper, otherwise, the characterisation tests are useless.

Response: More comparisons with literature have been made throughout Sections 3.1 and 3.2. 

Reviewer 3 Report

Sugarcane Bagasse and Orange Peels as Low Cost and Sustainable Bio-sorbents for the Removal of Lead (Pb(II)) From Contaminated Water (MS No. 1936720).

The topic should be corrected as “Sugarcane Bagasse and Orange Peels as Low Cost and Sustainable Bio-sorbents for the Removal of Lead ions in aqueous and Contaminated Water samples.

The manuscript entitled “Sugarcane Bagasse and Orange Peels as Low Cost and Sustainable Bio-sorbents for the Removal of Lead (Pb(II)) From Contaminated Water” is a scientific research paper focused on the removal of lead ions in aqueous and contaminated water samples using sugarcane bagasse and orange peels as adsorbents. The manuscript is designed appropriately but needs revision in the light of following comments/feedback.

1.      Line 98: write the correct formula of lead nitrate. Specify the symbol after lead nitrate.

2.      Line 108: Correct as C2H2.

3.      Line 141: After 5, clarify the symbol.

4.      Line 191: write the correct formula of lead nitrate. Explain the preparation of lead nitrate, write about the amount of lead nitrate (g) dissolved in 100 mL of distilled water.  

5.      Line 205: Write the correct formula of nitric acid.

6.      (2.6.2) Removal of lead ions using a combination of homogenized SCB and OPS biosorbents should be deleted. The studies related with the said sub-topic should also be deleted (dropped) from the experimental section as well as from results section.

7.      Line 268: Have you checked borehole water for lead ions and also the pH? If the water sample is devoid of lead ions, then consider/look for another water samples having lead ions (do not add lead ions from outside) and perform analysis at pH 5 and 4 accordingly as discussed. Based on the initial concentration of lead ions in real water samples and removal of lead ions after adsorption, calculate % removal of lead ions and discuss the effectiveness of the adsorption model.

8.      Line 446: Evaluation of pH effect on adsorption: The acidic pH (1-6) of lead ions is having an important role in the removal of lead ions. The reference details (likely to be cited) are given below: Journal Hazardous Materials 154 (2008) 407-416. The graph for the dependency of pH is on page 411. Even though, your result at pH 5 (in the pH range of 3 to 6) showed maximum removal of lead ions by SCB. Similarly, with OPS, the maximum removal of lead ions was at 4 (range 3 to 5). So, in the manuscript wherever, you have taken pH7, could not be accepted. So, change it with pH 5 and 4 accordingly. In all figures (Fig. 8,9,10), just focus on % removal of lead ions. Other figures such as (Fig. 8b,9b,10b) should be deleted, even delete (Fig. 8b,9b,10b) discussion.

9.      3.3.1.4. Evaluation of initial concentration ………………….

Just focus on metal uptake of lead ions at different concentration of lead ions, and compile a table based on the results obtained through Langmuir and Freundlich equations. For information in the relevant filed, follow the reference (likely to be cited) for said equations, details are given below: Journal of King Saud University – Science 32 (2020) 2931–2938. No need for figure 11a and b, delete them.

Author Response

Reviewer 3

  • The topic should be corrected as “Sugarcane Bagasse and Orange Peels as Low Cost and Sustainable Bio-sorbents for the Removal of Lead ions in aqueous and Contaminated Water samples.

Response: The correction has been made on the title of the manuscript. However, we have replaced “sustainable” with “reusable” as a result of comments made by the first reviewer.

The manuscript entitled “Sugarcane Bagasse and Orange Peels as Low Cost and Sustainable Bio-sorbents for the Removal of Lead (Pb(II)) From Contaminated Water” is a scientific research paper focused on the removal of lead ions in aqueous and contaminated water samples using sugarcane bagasse and orange peels as adsorbents. The manuscript is designed appropriately but needs revision in the light of following comments/feedback.

  • Line 98: write the correct formula of lead nitrate. Specify the symbol after lead nitrate.

Response: The symbol has been corrected.

  • Line 141: After 5, clarify the symbol.

Response: It has been corrected as 5 µm.

  • Line 191: write the correct formula of lead nitrate. Explain the preparation of lead nitrate, write about the amount of lead nitrate (g) dissolved in 100 mL of distilled water.

Response: The correct formula was used. The correction on the amount of lead nitrate (g) added in 100 mL of deionised water for the preparation of the 1000 mg/L Pb(II) solution was included in section 2.5.1.

  • Line 205: Write the correct formula of nitric acid.

Response: The correction has been made. Similar corrections have been carried out in other sections by formatting subscripts.

  • (2.6.2) Removal of lead ions using a combination of homogenized SCB and OPS biosorbents should be deleted. The studies related with the said sub-topic should also be deleted (dropped) from the experimental section as well as from results section.

Response: The effect of homogenised SCB and OPS was investigated to evaluate and compare the performance of the adsorbents for the removal of Pb(II) ions. The study was also investigated because we believe homogenised SCB and OPS may serve as a suitable alternative in the case that there is a shortage of any of the adsorbents due to seasonal variations. Thus, we haven’t deleted section 2.6.2. But we combined results and discussion section as recommended by reviewer 2.

  • Line 268: Have you checked borehole water for lead ions and also the pH? If the water sample is devoid of lead ions, then consider/look for another water samples having lead ions (do not add lead ions from outside) and perform analysis at pH 5 and 4 accordingly as discussed. Based on the initial concentration of lead ions in real water samples and removal of lead ions after adsorption, calculate % removal of lead ions and discuss the effectiveness of the adsorption model.

Response: The borehole water was checked for Pb(II) ions concentration and pH of water too. The Pb(II) concentration was found to be below the limit of detection of the method. The purpose of carrying out investigation using borehole water was to study the effect of the sample matrix in addition to simulated aqueous solutions. Therefore, spiking the borehole water samples enabled the assessment of the performance of the sorbent material in the presence of competitor ions such as Na(I), Ca(II), and Mg(II). Spiking water samples is accepted method in analytical chemistry and enables assessment of sample matrix. The pH of the borehole samples was adjusted to the aforementioned optimum pH before batch adsorption studies. The optimum pH for the removal of the Pb(II) ions was obtained at pH 7 for both biosorbents, therefore all adsorption studies that followed were conducted at pH 7.

  • Line 446: Evaluation of pH effect on adsorption: The acidic pH (1-6) of lead ions is having an important role in the removal of lead ions. The reference details (likely to be cited) are given below: Journal Hazardous Materials 154 (2008) 407-416. The graph for the dependency of pH is on page 411. Even though, your result at pH 5 (in the pH range of 3 to 6) showed maximum removal of lead ions by SCB. Similarly, with OPS, the maximum removal of lead ions was at 4 (range 3 to 5). So, in the manuscript wherever, you have taken pH7, could not be accepted. So, change it with pH 5 and 4 accordingly. In all figures (Fig. 8,9,10), just focus on % removal of lead ions. Other figures such as (Fig. 8b,9b,10b) should be deleted, even delete (Fig. 8b,9b,10b) discussion.

Response: The reported study in “Journal of Hazardous Materials 154 (2008) 407-416” was focused on “Removal of Pb(II) from aqueous solution by oxidized multiwalled carbon nanotubes”. But our study was focused on investigating the efficacy of green adsorbents (sugarcane bagasse and orange peels). We don’t expect the same adsorption behaviour for oxidized multiwalled carbon nanotubes and biosorbents (SCB and OPS). In our study, the optimum pH for the removal of the Pb(II) ions was found to be pH 7 for both biosorbents. Therefore, all adsorption studies that followed were conducted at pH 7. We haven’t deleted the suggested figures since reporting adsorption capacity is important in adsorption studies.

  • 3.1.4. Evaluation of initial concentration ...................... Just focus on metal uptake of lead ions at different concentration of lead ions, and compile a table based on the results obtained through Langmuir and Freundlich equations. For information in the relevant filed, follow the reference (likely to be cited) for said equations, details are given below: Journal of King Saud University – Science 32 (2020) 2931–2938. No need for figure 11a and b, delete them.

Response: Evaluation of the effect of initial metal ion concentration in adsorption studies is imperative. Thus, we haven’t deleted the suggested figures. Regarding the suggestion about adsorption isotherms and kinetics model, we have included as recommendation for future studies in Conclusions and Recommendations section.

Round 2

Reviewer 1 Report

The comments by the reviewer have been largely addressed.

Author Response

No comments on revised version

Reviewer 3 Report

The manuscript is revised accordingly and now accepted in the present form.

Author Response

Accepted the revised form